# Correlations of power output fluctuations in an offshore wind farm using high-resolution SCADA data

Janna Kristina Seifert[1], Martin Kraft[1,], Martin Kühn[1,], and Laura J. Lukassen[1]

[1]ForWind, Institute of Physics, Carl von Ossietzky University Oldenburg, Küpkersweg 70, 26129 Oldenburg, Germany

**Correspondence:** Janna Kristina Seifert (janna.kristina.seifert@uol.de)

**Abstract.** Space-time correlations of power output fluctuations of wind turbine pairs provide information on the flow conditions within a wind farm and the interactions of wind turbines. Such information can play an essential role in controlling wind turbines and short-term load or power forecasting. However, the challenges of analysing correlations of power output fluctuations in a wind farm are the highly varying flow conditions. Here, we present an approach to investigate space-time correlations of power output fluctuations of streamwise-aligned wind turbine pairs based on high-resolution SCADA data. The proposed approach overcomes the challenge of spatially variable and temporally variable flow conditions within the wind farm. We analyse the influences of the different statistics of the power output of wind turbines on the correlations of power output fluctuations based on eight months of measurements from an offshore wind farm with 80 wind turbines. First, we asses the effect of the wind direction on the correlations of power output fluctuations of wind turbine pairs. We show that the correlations are highest for the streamwise-aligned wind turbine pairs and decrease when the mean wind direction changes its angle to be more perpendicular to the pair. Further, we show that the correlations for streamwise-aligned wind turbine pairs depend on the location of the wind turbines within the wind farm and on their inflow conditions (free stream or wake). Our primary result is that the standard deviations of the power output fluctuations and the normalised power difference of the wind turbines in a pair can characterise the correlations of power output fluctuations of streamwise-aligned wind turbine pairs. Further, we show that clustering can be used to identify different correlation curves. For this, we employ the data-driven $k$-means clustering algorithm to cluster the standard deviations of the power output fluctuations of the wind turbines and the normalised power difference of the wind turbines in a pair. Thereby, wind turbine pairs with similar power output fluctuation correlations are clustered independently from their location. With this, we account for the highly variable flow conditions inside a wind farm, which unpredictably influence the correlations.

## 1 Introduction

Wind energy continues to be a growing source of energy. In 2019, 15.4 GW of new wind power capacity was installed in Europe, with 24% thereof located offshore (Komusanac et al., 2020). Considering the offshore wind power in 2019, the capacity in Europe has increased by 3.627 GW and a total of 7 wind farms were fully connected to the grid. Due to the increased size of the newly installed wind farms, the average size of offshore wind farms rose to 621 MW (Ramírez et al., 2020).

With the continuously increasing share of wind energy in the grid, the challenge of handling this highly fluctuating energy source becomes more important, as discussed in Ren et al. (2017). To convert wind energy into electrical energy, wind turbines are installed generally in groups (wind farms) at onshore and offshore sites. Fluctuations in their power output result from environmental influences such as changes in wind speed or wind direction, influences from neighbouring wind turbines and their state of operation. These power output fluctuations create challenges regarding the grid stability and are therefore an important field of investigation, (cf. Sorensen et al., 2007; Bossuyt et al., 2017b).

Wind turbines within a wind farm are placed as efficiently as possible to achieve the maximum power output for a respective site. The spacing of wind turbines is determined by the terrain of the site and the influence of wind turbines on each other (their wake). Wakes cause energy losses through reduced wind speeds and, at the same time, greater power output fluctuations and loads through increased turbulence (Crespo and Hernàndez, 1996; Vermeer et al., 2003).

Wake and wind farm flow effects on different spatial and temporal scales are reviewed by Porté-Agel et al. (2020). Many studies do not consider the power output fluctuations of wind turbines, which significantly impact the power output of a wind farm and the electrical grid. Thus, for further improvement of wind turbine control strategies like active power control (Vali et al., 2019) and grid stability by minute-scale prediction of offshore wind farm power (Valldecabres et al., 2020), the occurrence of power output fluctuations of wind turbines and their correlation within a wind farm are of great interest.

Andersen et al. (2017) investigated the influence of large coherent structures on the power output of wind turbines in large wind farms. The large coherent structures were found to cause high correlations in the power output of streamwise-aligned wind turbines. Research on wind speed correlations and power output correlations has shown that the wind turbines within a wind farm influence each other's power output fluctuations. Bossuyt et al. (2017a) found significant correlations of the power output for a streamwise-aligned set up of a wind farm of 100 porous disc models in a wind tunnel. Next to an increased turbulence intensity throughout the wind farm, the correlation of the power output reduced with the increasing distance of the discs. In an LES study by Lukassen et al. (2018), space-time correlations of velocity fluctuations within a wind farm with periodic boundary conditions (modelling a periodic array of wind turbines) were analysed and modelled analytically. Velocity fluctuations, which are directly related to power output fluctuations, showed pronounced space-time correlations. Furthermore, the variance of the wind velocity and the mean velocity turned out to be important parameters in the space-time correlation model. Stevens and Meneveau (2014) investigated the spectra of power output fluctuations of wind turbines in LES of finite-sized and infinitely large wind farms. The spectra were found to be dependent on the power output correlations of streamwise placed wind turbines. The power output correlation of the two wind turbines was significantly influenced by the wind direction, i.e. the lowest correlation for spanwise-placed wind turbines and highest correlation for streamwise-aligned wind turbines. Dai et al. (2017) analysed 1 Hz wind farm SCADA data concerning the influence of wind speed fluctuations around a mean velocity and wind direction fluctuations around a mean wind direction on the wind turbine power output fluctuations of single wind turbines. They showed a direct relation between the wind speed fluctuations and power output fluctuations in the partial load regime. Using 10-minute averaged wind farm SCADA data, Braun et al. (2020) derived a stochastic model for the power time series of wind turbines based on the temporal autocorrelation of the power of single wind turbines.

This work analyses 1 Hz wind farm SCADA data to describe the space-time correlations of the power output fluctuations of

wind turbine pairs. In contrast to the wind tunnel measurements by Bossuyt et al. (2017a) and the LES analysis by Lukassen et al. (2018) mentioned above, the data set processed here includes unstable inflow conditions (varying wind speeds and wind directions), dynamically operating wind turbines as well as changing flow conditions within the wind farm. Furthermore, there may be potential measurement inaccuracies. The result is a large and highly complex data set. In this paper, we investigate the influencing factors on the correlation of power output fluctuations of wind turbine pairs and introduce parameters to distinguish different correlation curves, herein called correlation states. A state defines a group of similar correlation curves. Note that the states found here refer to this specific wind farm and the considered time period. The parameters introduced to characterise correlation curves are then evaluated with a data-driven clustering algorithm to group the data according to the underlying correlation curves.

Starting with the description of the evaluated data set in Sect. 2, the processing of the data is explained in Sect. 2.1 and 2.2. The space-time correlation of power output fluctuations per wind turbine pair for time intervals of $600$ s is introduced. The correlation of wind turbine pairs is analysed in Sect. 3.1 for different wind directions using a filtered data set with less varying flow conditions. The correlation for wind directions with streamwise-aligned wind turbines is evaluated in more detail. In Sect. 3.2, the location dependence of the power output fluctuation correlation is determined by the comparison of wind turbine pairs located in different wind farm rows to confirm the findings of the wind tunnel measurements by Bossuyt et al. (2017a). With this and the results from the LES analysis by Lukassen et al. (2018), we identify relevant wind turbine power output statistics that influence the correlation. In Sect. 4, we use the straightforward $k$-means clustering approach (Lloyd, 1982) to group the data with respect to these statistical quantities, which show that they are clearly distinguishable correlation states. The conclusion and an outlook are drawn in Sect. 5.

## 2 Reference wind farm and data processing

The analysis performed in this work is based on measurements from the offshore wind farm Global Tech I (GT I). It is located in the North Sea, which is more than $100$ km off the coast of Northern Germany. Its total capacity of $400$ MW is provided by 80 wind turbines spread over an area of about $41$ km$^2$. The wind turbines of type Adwen AD 5-116 have a rated power of 5 MW, a rated wind speed of $12.5$ ms$^{-1}$, a hub height of $92$ m and a rotor diameter (D) of $116$ m. They are installed in a grid-like, slightly asymmetric pattern with a triangular shape towards the south (see Fig. 1).

The analysed data set was measured in about eight months, from January 1st, 2019 until September 9th, 2019 and consists of 1 Hz wind turbine SCADA data. The processed signals include the generated power $P$, the azimuth angle of the wind turbines (i.e. the nacelle direction) $\theta$, the nacelle-based wind direction $\varphi$ (measured relative to $\theta$), the pitch angle $\beta$ of each blade and a reconstructed wind speed $U$.

The reconstructed wind speed $U$ is not directly measured but provided as a variable that results from the measured power and control variables of the wind turbine (details on the reconstruction of $U$ are not available). Due to this, $U$ is considered as an approximated and idealised value that does not include the wind speed independent power reduction, e.g. by a yaw misalignment of the wind turbine. In this work, it can still be used to assess the effect of the wind speed on the correlations of

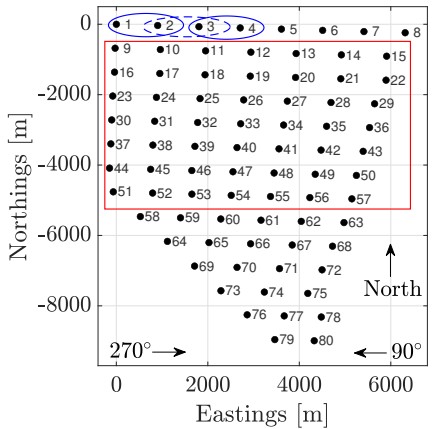

**Figure 1.** Layout of GT I. Each wind turbine is labelled with its corresponding number. The spacing of the wind turbines is inhomogeneous. The wind directions 90° and 270° (marked in the figure) will be analysed in detail in subsequent sections. The red square depicts the set of wind turbines that will later be used during the location-dependence analysis in Sect. 3.2 due to their symmetric arrangement. In the clustering analysis in Sect. 4, the whole wind farm will be used. The blue ellipses exemplarily show the definition of the considered wind turbine pairs. Tab. A1 lists the definition of all pairs.

power output fluctuations of wind turbine pairs, which will be further discussed in Sect. 2.2.

The azimuth angle $\theta$ of the wind turbine refers to the direction it is facing in its preset reference system. This system does not necessarily match the global geographical one due to the measurement inaccuracies of the azimuth angle and a potentially inaccurate north orientation of the reference system of each wind turbine (cf. Bromm et al., 2018).

    The nacelle based wind direction $\varphi$ is estimated based on the measurements of two 2D sonic anemometers installed behind the rotor of each wind turbine. These measurements have to be treated with care as the measured flow behind the rotor is disturbed

by the rotation of the rotor and the nacelle itself. Thus, it is only an estimation of the wind direction and yaw of the wind turbine. However, as shown by Dai et al. (2017), wind direction fluctuations at reasonable yaw angles ($< 45°$) have only little effect on the power output fluctuations of wind turbines. Thus, inaccuracies in $\varphi$ have no major influence on the performed analysis. The combined measurements of $\theta_i$ and $\varphi_i$ define the wind direction $\Phi_i$ at the $i$-th wind turbine.

    To assess the average wind direction for the wind farm, we average over $\Phi_i$ of all wind turbines to reduce the influence of false

measurements of single wind turbines. Due to the size of the considered wind farm, the wind direction is not expected to be consistent throughout the wind farm. Single wind turbines could be facing different wind directions compared to the average wind direction of the wind farm (cf. Schneemann et al., 2020; Sanchez Gomez and Lundquist, 2020). The wind direction of the wind farm that is averaged over all available wind turbines is defined as $\Phi_{av}$.

## 2.1 Data selection and filtering

Wind tunnel experiments and LES simulations as described in Sect. 1 pose controllable conditions for evaluating correlations. Such conditions cannot be met in a free-field wind farm. Next to temporally and spatially varying wind conditions, the wind farm layout leads to unequal conditions for wind turbines due to their positions, e.g. changing wind direction throughout the wind farm and asymmetric wind turbine spacing, especially for large wind farms. Further, each wind turbine operates independently from other wind turbines including yawing, pitching, starting up or shutting down. Next, single wind turbines

can be set to operate in a down-rated state or be shut down due to maintenance or other reasons. The combination of all these factors causes highly dynamic flow conditions and thus, an unpredictable variability. To cope with these issues, the data set is filtered for different operation states creating a cleaned data set with comparable operation state conditions for all wind turbines. For each considered interval of $600\,$s, the conditions defined in the following have to be met, cf. Tab. 1.

In general, a wind turbine operating in partial load is not pitching, and the velocity in its wake is always below the rated wind

speed. A wind turbine operating in full load aims at keeping a constant rotor speed and power by pitching its blades, where the wind speed in its wake can be larger than the rated wind speed. The data set is limited to partial load to avoid the effects of pitching and the different wake behaviour on the correlation. To further avoid effects from the transition from idle mode into operation or the transition from partial load to full load, only the data of the wind turbines generating power in the range of $0.5\,$MW and $4.5\,$MW is considered.

The previously defined limited power range still includes derated wind turbines. For derating wind turbines, their controller is manually changed so that their maximum power is limited to a certain value, which is lower than their nominal power. Due to this, wind turbines might start pitching already in the previously defined load range as their newly set power limit is already reached at lower wind speeds. Hence, to entirely exclude pitching wind turbines, the data is filtered for any pitching activity. Please note that for this specific data set, this implies that $\beta < -1.3°$.

Furthermore, yawing wind turbines are excluded from the analysis as well. The adjustment of wind turbines to the wind direction is managed by each wind turbine individually. Thus, wind turbines could be facing slightly different wind directions $\Phi_i$ and start yawing at different times. The yawing activity of a wind turbine transfers to its wake i.e. changes its deflection (cf. Bromm et al., 2018). Thus, yawing would affect the correlation for a pair of two wind turbines. To exclude yawing wind turbines, no change of $\theta$ is allowed in the regarded $600\,$s time interval: $\theta = const$.

To further filter the data for wind directions, the average wind direction $\Phi_{av}$ of all wind turbines is calculated for each time step of the regarded $600\,$s time interval. The average wind direction $\Phi_{av}$ has to fit the wind direction of interest within a

**Table 1.** Filters applied to the raw data of each wind turbine within the wind farm.

| Signal | Power | Pitch | Yaw |
|---|---|---|---|
| Settings | $0.5\,\text{MW} \leq P \leq 4.5\,\text{MW}$ | $\beta < -1.3°$ | $\theta = const.$ |

tolerance of $\pm 10°$ for all time steps in the regarded 600 s. Note that the borders of the interval include the lower limit and exclude the upper limit. Since this data filter only applies to the average wind direction $\Phi_{av}$, individual wind turbines might have a slightly deviating relative wind direction for this specific time interval. This deviation could be caused by a false wind direction measurement, a yawing process that has taken place asynchronously to the majority of other wind turbines or a wind direction deviation due to local changes over the area of the wind farm. This means there is no threshold for yaw misalignment within the 600 s intervals. As mentioned before, these effects have a limited effect on the power output fluctuations of the wind turbines.

As a summary, the overall data filtering procedure is as follows. The correlation analysis uses each time interval of 600 consecutive seconds where the two wind turbines of a wind turbine pair (as defined in Fig. 1) both pass all of the above-described data filters, i.e. power range, pitch, yawing and wind direction. This means that for different time intervals, a different set of wind turbine pairs is considered. Furthermore, wind turbine pairs can be considered for multiple time intervals.

## 2.2 Correlation of power output fluctuations

Power output fluctuations of individual wind turbines are defined as deviations of the instantaneous power from the average power of the regarded wind turbine $i$ within a certain time interval $\Delta t$. We analyse time intervals of $\Delta t_{600} = 600$ s:

$$P'_{i,t_j}(t) = P_i(t) - \langle P_i(t) \rangle_{\Delta t_{600}} \tag{1}$$

where $\langle P_i(t) \rangle_{\Delta t_{600}}$ is the average of the measured power $P_i(t)$ over an interval $\Delta t_{600}$, including all 600 values for $t$ in the discretised interval $[t_j, t_j + 599 \text{ s}]$. $P'_{i,t_j}(t)$ is the power output fluctuation within the interval $\Delta t_{600}$ (the index $t_j$ is omitted in the following). Depending on the data availability, the next interval of 600 consecutive seconds could go from $[t_j + 1 \text{ s}, t_j + 1 \text{ s} + 599 \text{ s}]$, and thus overlap the previous one up to 599 s. This does not result in significantly different findings compared to non-overlapping intervals as shown in App. B.

The selection of the interval size of 600 s is based on the layout of the wind farm and the considered power ranges or corresponding wind speeds. For example, considering the spacing of up to 9 D for westerly winds, with a cut-in wind speed of $4 \text{ ms}^{-1}$ and a rated wind speed of $12.5 \text{ ms}^{-1}$, a particle moving with the undisturbed wind would take from about 84 s up to 261 s to travel from one wind turbine to its downstream neighbour. Taking a lower advection wind speed within the wind farm into account, a considered interval length of 600 s captures potential correlations of interest. Each time step followed by 599 consecutive time steps forms an interval, individually for each wind turbine. For all available intervals of all wind turbines, the power output fluctuations are calculated based on Eq. 1.

To analyse the influence of wind turbines on each other, the space-time correlation is calculated using the Pearson correlation coefficient (Pearson, 1896)

$$r(\tau) = \frac{\langle P'_A(t) P'_B(t+\tau) \rangle_{\Delta t_{300}}}{\sqrt{\langle P'^2_A(t) \rangle_{\Delta t_{300}} \langle P'^2_B(t+\tau) \rangle_{\Delta t_{300}}}} \tag{2}$$

where $\langle ... \rangle_{\Delta t_{300}}$ is the average over an interval $\Delta t_{300} = 300$ s including all 300 values for $t$ in the discretised interval $[t_j, t_j + 299 \text{ s}]$, $r(\tau)$ is the Pearson correlation coefficient in dependence of a time lag $\tau$, $P'_A(t)$ is the power output fluctu-

ation of the upstream wind turbine A following Eq. 1 at a time $t$, $P'_B(t+\tau)$ is the power output fluctuation of the downstream wind turbine B at a time $t+\tau$ with a time lag $\tau$.

The Pearson correlation coefficient is a value between -1 and 1, where 1 depicts the maximum possible linear correlation, -1 is the maximal linear anti-correlation and a value of 0 depicts no linear correlation. The correlation coefficient is evaluated for a fixed period of 300 s from $P'_A(t)$ to $P'_A(t+300\,\text{s})$ and likewise $P'_B(t+\tau)$ to $P'_B(t+300\,\text{s}+\tau)$. This allows a maximum time lag of $\tau = 300$ s for each considered 600 s interval.

Similar to Taylor's hypothesis (Taylor, 1938), we assume that the wind structures responsible for the power output fluctuations measured at an upstream wind turbine A that travel a certain distance to the downstream wind turbine B with an advection speed that is similar to the average wind speed over that distance. But in contrast to Taylor's hypothesis, we do not assume frozen eddies but expect wind structures to change and thus decorrelate while travelling downstream. Further, as we have no access to the average wind speed over the distance between wind turbines A and B, we use the average wind speed measured at wind turbine B as a reference. Hence, to compare the correlations calculated for intervals with different average wind speeds and different wind turbine distances, the time lag $\tau$ is normalised for each time interval starting at $t_j$ individually

$$\tau_{norm,intv} = \tau \cdot \frac{\langle U_B(t+\tau)\rangle_{\Delta t_{300}}}{x_{AB}} \tag{3}$$

where $\tau_{norm,intv}$ is the normalised time lag, $\langle U_B(t+\tau)\rangle_{\Delta t_{300}}$ is the average reconstructed wind speed from a certain (downstream) wind turbine B for a time interval $\Delta t_{300} = 300$ s for $t$ in the discretised interval $[t_j, t_j + 299\,\text{s}]$ and a certain lag $\tau$. This means for a certain $\tau$, the averaging interval of $\langle U_B(t+\tau)\rangle_{\Delta t_{300}}$ is $[t_j + \tau, t_j + \tau + 299\,\text{s}]$. $x_{AB}$ is the distance between wind turbine A and wind turbine B.

Next, the correlation curves with the normalised lag $\tau_{norm,intv}$ are discretised using a histogram with a reference time lag of

$$\tau_{norm} = \tau \cdot \frac{U_{max}}{x_{AB,mean}} \tag{4}$$

where $\tau$ is the time lag (0 s to 300 s), $U_{max}$ is an artificially introduced velocity that has to be at least equal to the maximum possible wind speed to fit all normalised curves ($U_{max} = 13\ \text{ms}^{-1}$ for this case). This value is based on the wind turbine power curve characteristics, including a tolerance as the wind turbines considered here reach their rated power at $12.5\ \text{ms}^{-1}$. $x_{AB,mean}$ is the average distance between wind turbine A and wind turbine B of the considered wind turbine pairs. Note that $\tau_{norm,intv}$ is used for stretching and shrinking of the correlation curves. $\tau_{norm}$ is only a reference time lag that is only created for binning of the stretched or shrunk correlations and does not change the correlation curves.

Due to the definition of $\tau_{norm,intv}$ and $\tau_{norm}$ (see Eq. 3 and 4), the peak of the correlation curves is expected to be found around $\tau_{norm,intv} = 1$ if the advection speed of the wind speed fluctuations matches the wind speed affecting wind turbine B. Thus, in partial load situations where wind turbine B is in the wake of wind turbine A, the peak is expected to be at $\tau_{norm} > 1$. Here, the reduced wind speed in the wake recovers slowly, so that the wind speed affecting wind turbine B, i.e., $U_B$ is already partly recovered and hence larger than the advection speed of the fluctuations. As mentioned before, $U_B$ is reconstructed and might differ from the actual wind speed affecting the wind turbines. However, in the context of this normalisation, the effect on the resulting correlations curves is marginal as the correlation curves may only be slightly shifted due to the deviation from the real wind speed.

## 3   Wind direction dependence and location dependence

As described in Sect. 1, this work aims to study the influences of the free-field wind farm situation on the space-time correlations
of the power output fluctuations. For this, we analyse the time intervals of a fixed set of 66 wind turbine pairs, namely those
that are streamwise-aligned for the wind directions 90° and 270° (see Fig. 1 and Tab. A1). Note that the pairs are the same for
both wind directions, but the order of the evaluation for the wind direction-dependent correlation differs (i.e. the upstream and
downstream wind turbine position of a pair is reversed).

In the following, we average correlations over a wind direction interval of 20° and all available time intervals of the considered
wind turbines (either all wind turbines or a selection of wind turbines). We consider 20° intervals due to a 10° tolerance in the
wind direction measurements of the wind turbines. The averaged correlation is denoted by $R(\tau_{norm})$. In Sect. 3.1, the average
correlation for the 66 wind turbine pairs is analysed for each wind direction interval separately. Further, the location-dependent
correlations for the wind turbine pairs are evaluated, and wind turbine statistics that characterise the power output fluctuation
correlations are investigated in Sect. 3.2.

### 3.1   Wind direction-dependent space-time correlation

After applying the data filters described in Sect. 2.1 to the intervals of the 66 wind turbine pairs, the average correlation per
wind direction is determined. For each wind turbine pair, the power output fluctuation correlations are averaged over the wind
direction intervals of 20°, which are applied to steps of 10°, i.e. the interval for the wind direction 90° corresponds to the
directions $80° \leq \Phi_{av} < 100°$ and the consecutive interval for the wind direction 100° is $90° \leq \Phi_{av} < 110°$. For the 10°-wind
direction steps from 0° to 170°, we treat the pairs according to Tab. A with a reverse order and for the 10°-wind steps from 180°
to 350°, we treat the pairs with the given order. This means that even for the wind directions where both the wind turbines of
a pair are parallel to the wind direction, the 'upstream' wind turbine A is chosen according to the table. Afterward, the results
are averaged over all wind turbine pairs for each 10° step separately. Due to the different availability of each wind turbine pair,
they influence the average correlation to a different amount. Figure 2 displays the amount of data of all correlation intervals
of all wind turbine pairs per wind direction interval of 20°. The main wind direction is about 220° and shows the maximum
occurrence, whereas 90° has about 20% fewer data and 270° has about 45% lesser data. For wind directions from 350° to 20°,
there was almost no data available.

 Figure 3 displays the averaged power output fluctuation correlation per 10° wind direction step, which is averaged over the
20° wind direction interval and all the time intervals of all available wind turbine pairs. The averaged correlation coefficient
is plotted as colour and the time lag $\tau_{norm}$ (Eq. 4) is denoted as the radius. Due to the varying data availability per wind
direction and the applied data filtering (see sec. 2.1), the average correlation curve per bin is based on a different number of
data. It turns out that after filtering, no data is available for the bins around 330° to 10°. Wind turbines in a pair are streamwise-
aligned for wind directions around 90° and 270°. Fluctuations take a certain time to travel from one wind turbine to the other,
where the fluctuations are influenced by the upstream wind turbine. The highest correlation peak is at $\tau_{norm} > 1$ according
to the definition of $\tau_{norm}$ in Eq. 4. At 90°, a correlation of about 0.16 is found, whereas, a correlation of about 0.2 is noted

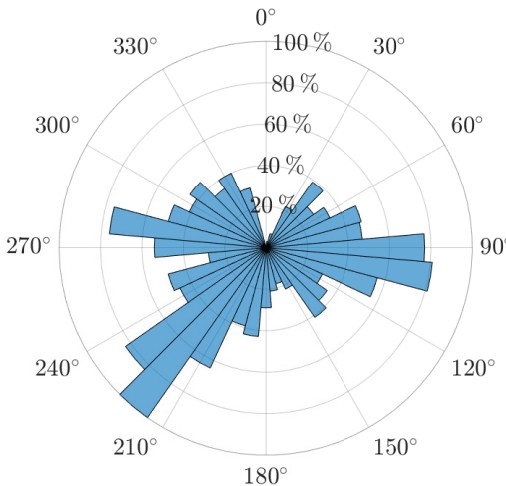

**Figure 2.** Availability of data per wind direction interval normalised to the number of the available correlation intervals for 220°, namely 9,102,133 intervals. The outer labels depict the wind direction and the inner circles denote the percentage of availability.

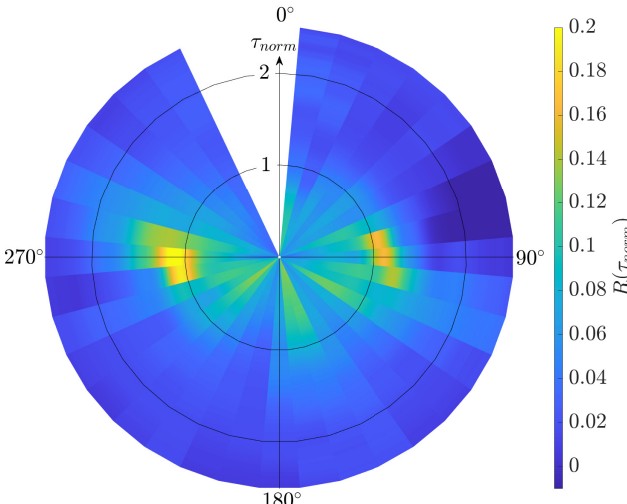

**Figure 3.** Average power output fluctuation correlation per 10° wind direction step of all available wind turbine pairs within the wind farm. Since the power output fluctuation correlation is averaged for wind direction intervals of 20° the intervals are overlapping by 10°. For a better visibility the intervals are visualised in 10° steps only, i.e. for 90° the interval goes from 80° to 100° but is visualised from 85° to 95°. The radius of the circle is the time lag $\tau_{norm}$, i.e. $\tau_{norm} = 0$ is in the origin and $\tau_{norm} = 1$ is on the inner black circle. No data was available for the wind direction interval around 350° (cf. Fig. 2).

at 270°. The maximum correlation around 0.2 may seem relatively low but it is reasonable considering the high variability in the flow and wind turbine dynamics in free-field measurements. These dynamics are most likely caused by the wind direction

and wind speed changes and the individual operation of the wind turbines (yawing, limited power, shut off). Even though the correlation curves were adapted to the average wind speed per interval, the wind speeds were just an assessment and could change during the interval. Also, the wind direction is averaged over the whole wind farm, which means certain intervals could include data from wind turbine pairs facing a slightly different direction. Further, we only consider the intervals of wind turbine pairs that fit the data filter; however, other wind turbines could be yawing at the same time or start pitching. Thus, the flow within the wind farm could still be influenced by these wind turbines. In the LES study of Lukassen et al. (2018), a maximum correlation coefficient of about 0.5 was found for the space-time correlations of wind speeds measured at comparable distances with comparable wind speed. In the wind tunnel experiments by Bossuyt et al. (2017a), a maximum correlation of about 0.55 was found for the space-time correlation of the reconstructed power output of discs placed at comparable distances with comparable wind speeds. In both the simulations and experiments, the flow conditions are ideal compared to those in the free-field measurements. For wind directions approaching 0° and 180°, the wind turbines in a pair are oriented more perpendicular to the wind direction, and the fluctuations reach both wind turbines A and B at nearly the same time. This leads to a change in the expected position of the highest peak and the peak magnitude of the correlation curves. The found correlations are not as pronounced as those for the streamwise case (i.e. around 90° and 270°), which confirms the simulation results by Stevens and Meneveau (2014). Thus, we will not investigate the spanwise correlations in further detail.

Figure 4 shows the average power output fluctuation correlation around 90° and 270° in detail as cuts through Fig. 3. The absolute peaks are at 90° and 270°. For wind directions where the wind turbines in a pair are less streamwise-aligned, the peak decreases and the correlation curve flattens. The correlations for 270° are more defined and show slightly larger peak values compared to those for 90°. This may be due to the asymmetric wind farm layout (cf. Fig. 1). The deviation between the average correlation curves for wind directions around 260° and 280° could be caused by the not entirely horizontally aligned wind turbines and by the triangular shape at the lower part of the wind farm; however, this phenomenon is not further investigated in this analysis.

## 3.2 Location-dependent space-time correlation

The location dependence of the averaged power output fluctuation correlations is investigated for wind direction intervals around 90° and 270°. As mentioned before, the wind turbines are, on average, streamwise-aligned for these two wind direction intervals. The most northern wind turbines, 1 to 8, and wind turbines 58 to 80 in the lower triangle of the wind farm do not follow the symmetric pattern of the square consisting of wind turbines 9 to 57. Thus, the following results are limited to this symmetric square as marked in Fig. 1.

Figure 5 displays the averaged correlations of the power output fluctuations for all wind turbine pairs included in the upper square of the wind farm for the wind direction intervals 90° and 270°. A total of 4,916,277 intervals and 3,329,333 intervals of 600 s are averaged for 90° and 270°, respectively. Similar to Fig. 4, both correlation curves show similar shapes, whereas the correlation for 270° is generally higher than that for 90°. The maximum averaged correlation coefficient is about 0.16 and 0.21 for 90° and 270°, respectively.

Further, the power difference (normalised by the average power output of the upstream wind turbine A) and the average standard

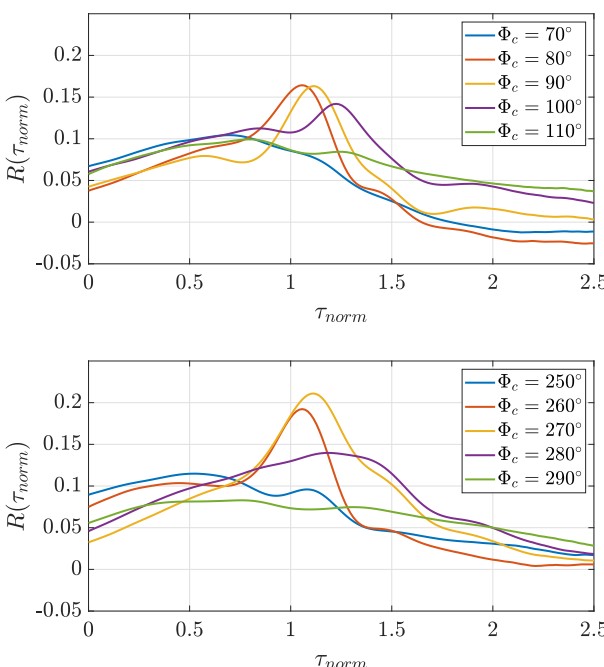

**Figure 4.** Average power output fluctuation correlations for wind direction intervals from around 70° to 110° and around 250° to 290° as radial cuts through Fig. 3. $\Phi_c$ depicts the centre of the wind direction intervals.

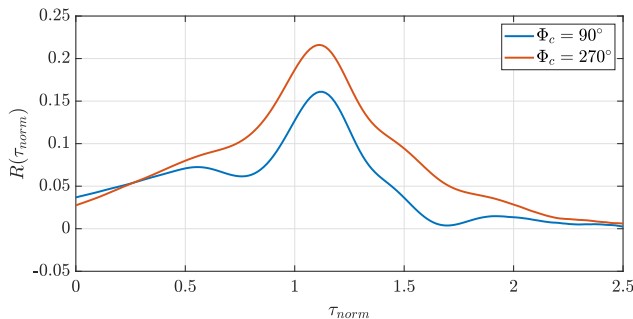

**Figure 5.** Average power output fluctuation correlation for wind direction intervals around 90° and 270° considering the wind turbines 9 to 57 in the symmetric square (cf. Fig. 1). $\Phi_c$ depicts the centre of the wind direction intervals.

deviation of the power output fluctuations of both wind turbines in a pair are determined to analyse the flow conditions. The results for 90° and 270° are listed in Tab. 2. For both wind directions, the averaged standard deviation of the power output fluctuations is larger for the downstream wind turbine B than for the upstream wind turbine A. However, the averaged standard deviation of the power output fluctuations for 90° is smaller than that for 270°. The normalised power difference of the wind turbine pairs for 90° and 270° is about 12% and 8%, respectively. The different behaviour is likely to be caused by the distinct meteorological conditions, e.g. distribution of mean wind speed and atmospheric stability, between the two wind directions.

**Table 2.** Averaged wind turbine statistics computed for the wind direction intervals around 90° and 270° with $A$ as the upstream wind turbine and $B$ as the downstream wind turbine. $\sqrt{\langle P_A'^2 \rangle_{\Delta t_{600}}}$ is the standard deviation of the power output fluctuations of wind turbine $A$ over 600 s intervals $\Delta t_{600}$ (analogue for wind turbine $B$ for the same 600 s intervals, respectively). $\langle P_A \rangle_{\Delta t_{600}}$ and $\langle P_B \rangle_{\Delta t_{600}}$ are the average power of wind turbines $A$ and $B$ over the same 600 s intervals. $\langle ... \rangle_{all}$ denotes the average of the statistics over all available time intervals of the wind turbine pairs. Note that $\Phi_c$ depicts the centre of 20° wind direction intervals, here from 80° to 100° and from 260° to 280°.

| $\Phi_c$ | $\left\langle \sqrt{\langle P_A'^2 \rangle_{\Delta t_{600}}} \right\rangle_{all}$ [kW] | $\left\langle \sqrt{\langle P_B'^2 \rangle_{\Delta t_{600}}} \right\rangle_{all}$ [kW] | $\left\langle \frac{\langle P_A \rangle_{\Delta t_{600}} - \langle P_B \rangle_{\Delta t_{600}}}{\langle P_A \rangle_{\Delta t_{600}}} \right\rangle_{all}$ |
|---|---|---|---|
| 90° | 212 | 222 | 0.12 |
| 270° | 247 | 260 | 0.08 |

To further investigate the wind turbine location dependence of the power output fluctuation correlations, the average correlation of wind turbine rows is calculated for both wind directions. Here, a wind turbine row consists of a line of wind turbines perpendicular to the incoming wind, as shown in the upper right corner of Fig. 6. For both wind directions, no sharp correlation is found for the first row (turbine A in the first row, turbine B downstream of A, dark blue curves). It should be noted that the upstream wind turbine $A$ is standing in the free stream, while the downstream wind turbine $B$ is affected by the wake of the upstream wind turbine. Thus, the two wind turbines have very different inflow conditions. For wind turbine pairs located further downstream, both wind turbines are standing in the wake of the upstream wind turbines. Here, a clear correlation is found. For the correlation curves of the second to last row, the peaks become more defined as their width decreases.

As described by Bossuyt et al. (2017a), the turbulence intensity increases with the flow towards the back of the wind farm. Furthermore, as indicated above in Lukassen et al. (2018), the variance of wind speed fluctuations plays an important role in modelling the velocity space-time correlations. To evaluate the row-dependent conditions in the measurement data, Tab. 3 lists the average standard deviations of the power output fluctuations measured at the upstream and downstream wind turbine of all pairs, as well as the average normalised power difference of all pairs. For both wind turbines in a pair, the average standard deviations of the power output fluctuations show a clear increasing trend throughout the wind farm, which is similar to that of the turbulence intensity in the wind tunnel results mentioned above. The normalised power difference is largest for the first row, which is caused by the previously described deviating inflow conditions of the upstream and downstream wind turbine. This was also found in the experiment by Bossuyt et al. (2017a) where the first row generates the maximum power, while the second and following rows show a significant reduction.

## 4 *K*-means clustering of power output fluctuation characteristics

Results of section 3.2 reveal that the standard deviation of the power output fluctuations and the power difference of the wind turbines change depending on the location of the wind turbine (pairs) within the wind farm. As explained in section 2.1, conditions in a wind farm are never ideal due to the variety of influence factors such as wind direction and wind speed

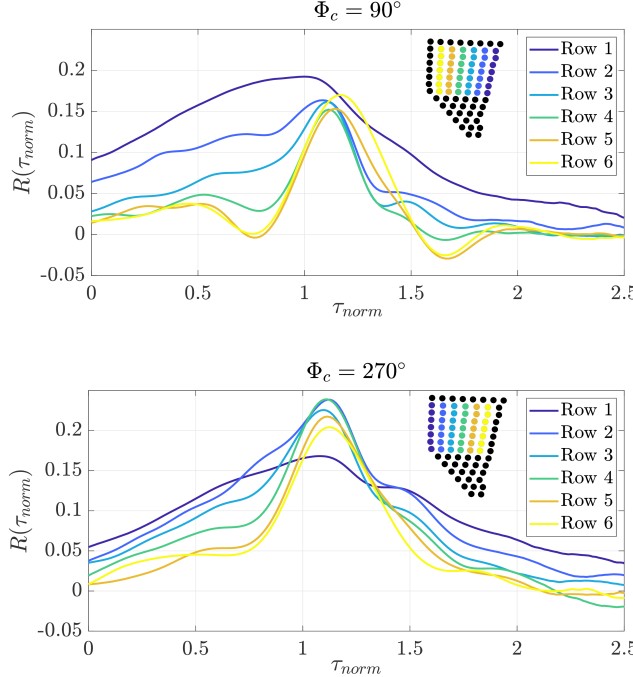

**Figure 6.** Averaged power output fluctuation correlation per wind farm row for wind direction intervals around 90° (top) and 270° (bottom) considering all wind turbines in the symmetric square (cf. Fig. 1). $\Phi_c$ depicts the centre of the wind direction intervals. For the considered correlation curves, wind turbine A is located in the respective coloured row and turbine B is one row downstream of A. As the wind turbines are analysed in pairs of two, the last row of wind turbines is unlabelled, as these wind turbines do not have a downstream partner. For both figures, the numbering and the colours of the rows are identical with regard to the considered wind direction.

fluctuations or influences of surrounding wind turbines. Turbines within the wind farm that are turned off or derated might create free stream like inflows for downstream wind turbine pairs. Such irregularities influence the standard deviations of the power outputs and the normalised power differences calculated for wind turbine pairs. For example, a wind turbine that is turned off for a certain time interval is not considered in the analysis. However, it still influences the flow conditions within the wind farm and the statistics or correlations of the surrounding wind turbine pairs. Thus, a considered wind turbine pair downstream of the non-operating wind turbine could show a different correlation than that if the upstream wind turbine would be turned on. To identify these locally abnormal conditions and the resulting deviations in the power output fluctuations and their correlations, the $k$-means clustering algorithm is used to sort the correlations based on previously defined statistics, standard deviation and the normalised power difference of the wind turbines in a pair.

$k$-means is an algorithm that iteratively sorts data into $k$ clusters. After choosing an initial centre for each cluster (centroids) within the data, all data points are assigned to their nearest centroid. Afterwards, the new centres of the clusters are calculated based on the assigned data points. These steps are repeated until a previously defined number of iterations is reached or when the centres of the clusters no longer change. Finally, the data is distributed into $k$ clusters. The result of the $k$-means algorithm

**Table 3.** Averaged wind turbine statistics per wind farm row computed for wind direction intervals around 90° and 270° with $A$ as the upstream wind turbine and $B$ as the downstream wind turbine. $\sqrt{\langle P_A'^2 \rangle_{\Delta t_{600}}}$ is the standard deviation of the power output fluctuations of wind turbine $A$ over a 600 s interval $\Delta t_{600}$ (analogue for wind turbine $B$ for the same 600 s intervals). $\langle P_A \rangle_{\Delta t_{600}}$ and $\langle P_B \rangle_{\Delta t_{600}}$ are the average power outputs of wind turbines $A$ and $B$ over the same 600 s intervals. $\langle ... \rangle_{row}$ denotes the average of the statistics over all available time intervals of the wind turbine pairs in a row. Note that 90° and 270° again refer to the 20° wind direction intervals from 80° to 100° and from 260° to 280°.

| | $\left\langle \sqrt{\langle P_A'^2 \rangle_{\Delta t_{600}}} \right\rangle_{row}$ [kW] | | $\left\langle \sqrt{\langle P_B'^2 \rangle_{\Delta t_{600}}} \right\rangle_{row}$ [kW] | | $\left\langle \frac{\langle P_A \rangle_{\Delta t_{600}} - \langle P_B \rangle_{\Delta t_{600}}}{\langle P_A \rangle_{\Delta t_{600}}} \right\rangle_{row}$ | |
|---|---|---|---|---|---|---|
| Row | 90° | 270° | 90° | 270° | 90° | 270° |
| 1 | 114 | 166 | 164 | 216 | 0.27 | 0.19 |
| 2 | 186 | 232 | 222 | 258 | 0.05 | 0.02 |
| 3 | 224 | 254 | 242 | 265 | 0.09 | 0.06 |
| 4 | 243 | 269 | 241 | 268 | 0.10 | 0.06 |
| 5 | 250 | 276 | 234 | 274 | 0.08 | 0.08 |
| 6 | 232 | 280 | 220 | 278 | 0.06 | 0.04 |

is dependent on the starting positions of the cluster centres. Thus, the algorithm can be repeated with changing starting points for the clusters to find the best possible solution.

In the following, we investigate the clustering results for the directions 90° and 270°. Here, the triangular shape of the lower part of the wind farm (wind turbines 58 to 80), and the most northern wind turbines 1 to 8 are now incorporated (cf. Fig. 1) to identify the flow conditions of the whole wind farm. This results in 6,985,830 considered time intervals for 90° and 4,914,448 considered time intervals for 270°. Clustering is performed using the $k$-means algorithm of MATLAB (MATLAB, 2019) based on Lloyd (1982) using random sample points as initial centroids to find the best solution. Clustering is repeated ten times to

avoid the generation of local centroids, and the run with the clusters with the lowest sum of point-to-centroid distances within the clusters is chosen. As a distance metric for the clusters, the squared Euclidean distance is chosen. The maximum number of iterations is set to 300 and $k$ is set to five clusters. This number was empirically chosen as the data was grouped into a reasonable set of groups with clearly distinguishable correlation curves (correlation states). A greater number of clusters lead to further clusters with similar correlation curves. The only difference found was in the standard deviation of the power output

fluctuations of the wind turbine pairs. Here, the cluster indicates a higher standard deviation of the power output fluctuations for the upstream wind turbine A instead of the downstream wind turbine B. This slightly abnormal behaviour is shown in more detail in Appendix C. Also, different orderings of the intervals have been tested, namely random sorting, data sorted for an increasing standard deviation of the power output fluctuations of the downstream wind turbine $B$, and chronological sorting according to the available time intervals. The results are equal, including the first decimal place of the centroids for all cases.

Thus, a random sorting is used in further analysis.

Table 4 lists the determined centroids (centres of the clusters) for wind directions 90° and 270°. Standard deviations of the power output fluctuations of both wind turbines $A$ and $B$ significantly decrease while the normalised power difference of $A$ and $B$ is significantly increasing from Cluster 1 to 5. To further investigate these findings, we analyse the correlation curves corresponding to the clusters. Figures 7 and 8 show the average correlations for both wind directions for each of the five clusters (upper plots) and the percentage frequency of each pair within each of the five clusters (lower plots). As expected from Fig. 4 and 5, the average correlations for 270° are higher than those for 90°. Cluster 1 includes nearly 6% of the data and has the highest correlation. This is a significant increase compared to the average correlation shown in Fig. 5. The correlation decreases while the amount of data per cluster increases from Cluster 2 to 4. No correlation is found for Cluster 5. A clear trend is visible upon looking at the occurrence of wind turbine pairs within each cluster. While Cluster 1 with the highest correlation is dominated by wind turbine pairs, where the upstream wind turbine is located towards the back of the wind farm, Cluster 5 with no correlation is dominated by wind turbine pairs with its upstream wind turbine located in the first row of the wind farm. From Clusters 2 to 4, the dominating wind turbine pairs shift from the back rows towards the front rows, whereas the percentage frequency becomes more balanced throughout the wind farm (i.e. more light green coloured turbines).

The comparison of the results of Fig. 7 and 8 and Tab. 4 depicts that the greater the standard deviations of the power output fluctuations and the smaller the normalised power difference of the wind turbines in a pair, the higher the correlation for´ the wind turbine pairs. The slight row dependence, which was already indicated in Tab. 3, can be confirmed here. This is illustrated by a colour coding of the frequency of occurrence of wind turbine pairs in each cluster in the lower subplot of Fig. 7 (respectively Fig. 8). The sum of all frequencies of all wind turbines within one cluster adds up to 100%, meaning a yellow coloured wind turbine pair makes up about 3% of the respective cluster, and a green-marked wind turbine pair makes up about 1.5% of the respective cluster. For example, the correlation peak for Cluster 1 of more than 0.3 for 90° (respectively

**Table 4.** Cluster centroids for wind direction intervals around 90° and 270° with $A$ as the upstream wind turbine and $B$ as the downstream wind turbine. $\sqrt{\langle P_A'^2 \rangle_{\Delta t_{600}}}$ is the standard deviation of the power output fluctuations of wind turbine $A$ over 600 s intervals $\Delta t_{600}$ (analogue for wind turbine $B$ for the same 600 s intervals, respectively). $\langle P_A \rangle_{\Delta t_{600}}$ and $\langle P_B \rangle_{\Delta t_{600}}$ are the average power output of wind turbines $A$ and $B$ over the same 600 s intervals. $\langle ... \rangle_{cluster}$ denotes the average of the statistics over all available time intervals of the wind turbine pairs within a cluster. Note that 90° and 270° again refer to 20° wind direction intervals from 80° to 100° and from 260° to 280°.

| | $\left\langle \sqrt{\langle P_A'^2 \rangle_{\Delta t_{600}}} \right\rangle_{cluster}$ [kW] | | $\left\langle \sqrt{\langle P_B'^2 \rangle_{\Delta t_{600}}} \right\rangle_{cluster}$ [kW] | | $\left\langle \frac{\langle P_A \rangle_{\Delta t_{600}} - \langle P_B \rangle_{\Delta t_{600}}}{\langle P_A \rangle_{\Delta t_{600}}} \right\rangle_{cluster}$ | |
|---|---|---|---|---|---|---|
| Cluster | 90° | 270° | 90° | 270° | 90° | 270° |
| 1 | 513 | 535 | 527 | 540 | 0.02 | 0.05 |
| 2 | 367 | 381 | 387 | 405 | 0.05 | 0.06 |
| 3 | 253 | 283 | 271 | 298 | 0.10 | 0.06 |
| 4 | 157 | 197 | 174 | 213 | 0.13 | 0.07 |
| 5 | 73 | 117 | 86 | 133 | 0.14 | 0.11 |

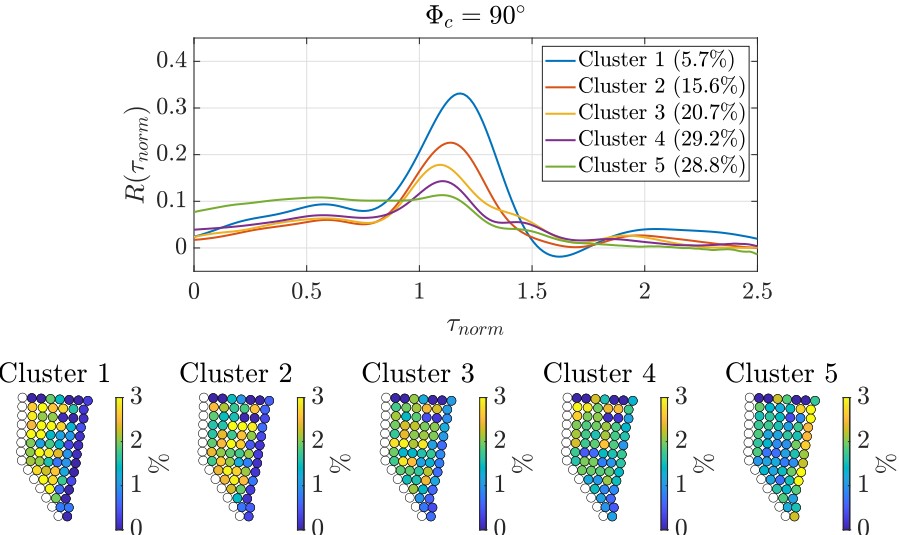

**Figure 7.** Clustering for wind direction interval around 90° with randomly sorted 600 s time intervals. $\Phi_c$ depicts the centre of the wind direction interval. The upper plot shows the average power output fluctuation correlation curve per cluster. The legend lists the share of data. The lower plot shows the percentage frequency of each wind turbine pair within the respective cluster given as colour. As the wind turbines are analysed in pairs of two, the last row of wind turbines is unlabelled, as these wind turbines do not have a downstream partner.

0.4 for 270°) partly includes pairs with the upstream turbine in the last row and some turbine pairs in the rows before. This is considerably larger than the correlation curve of row 6 of Fig. 6.

## 5    Conclusions

We presented an approach to analyse the correlations of power output fluctuations of wind turbine pairs for 600 s time intervals
based on 1 Hz SCADA data, which copes with the challenge of highly variable flow conditions in the measurement data and
the identification of correlation states. Further, we investigated different influences on the correlation of power output fluctua-
tions of wind turbine pairs. The investigation of the influence of different wind directions on the correlations of power output
fluctuations of wind turbine pairs showed that streamwise-aligned pairs are correlated. In contrast, spanwise pairs show nearly
no correlation of power output fluctuations. Thus, we focused our investigation on the streamwise wind turbine pairs.
Inspired by the findings of Bossuyt et al. (2017b), which showed an increasing turbulence intensity throughout the wind farm
and the model for velocity space-time correlations by Lukassen et al. (2018), we introduced and evaluated parameters to char-
acterise correlation states of power output fluctuations. The chosen parameters were the standard deviations of the power output
fluctuations and the normalised power difference of wind turbines in a pair.
In general, we found that the averaged correlation curves of power output fluctuations for 270° with a maximum correlation
coefficient of 0.21 have more defined (narrower) peaks compared to those of the averaged correlation curves for 90° with a

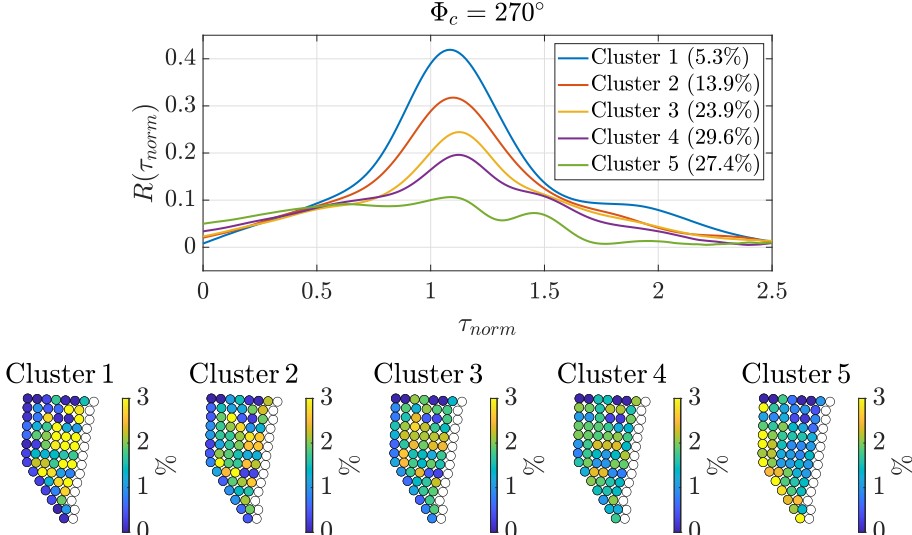

**Figure 8.** Clustering for wind direction interval around 270° with randomly sorted 600 s time intervals. $\Phi_c$ depicts the centre of the wind direction interval. The upper plot shows the average power output fluctuation correlation curve per cluster. The legend lists the share of data. Note that the values do not add up exactly to 100% due to rounding. The lower plot shows the percentage frequency of each wind turbine pair within the respective cluster given as colour. As the wind turbines are analysed in pairs of two, the last row of wind turbines is unlabelled, as these wind turbines do not have a downstream partner.

maximum correlation coefficient of 0.16. Further, the standard deviation of the power output fluctuations of the wind turbines in a pair was larger for 270° than for 90°. This difference, together with the slightly asymmetric layout of the wind farm and different inflow conditions for 90° and 270°, are most likely the root causes for this deviation in the correlation curves. In the context of the considered highly varying flow conditions, peak correlations around 0.21 or 0.16 are still considered significant.

The cause for these relatively low peak correlations lies in the varying flow conditions or noisiness of the flow within the wind farm.

The investigation of the average correlation for wind turbine pairs per wind farm row strengthened our previous findings. We found different correlation curves for the rows of the wind farm, becoming more defined (more narrow peaks) towards the back of the wind farm. Wind turbine pairs, where the upstream wind turbine A is located in the first row and the downstream wind

turbine B is located in the second row of the wind farm, show no correlation. In addition, large normalised power differences of the wind turbines in a pair and small standard deviations of power output fluctuations were observed. This is most likely caused by the free stream inflow of the upstream wind turbine A of the pairs. Most importantly, the analysis of the separate rows of the wind farm revealed a trend of increasing standard deviations of the power output fluctuations throughout the wind farm and a decreasing normalised power difference of the wind turbines in a pair. As mentioned before, the flow throughout

the wind farm is highly variable due to the individual control and operation of the wind turbines. This means that not all wind turbine pairs in the same row are affected by the same flow conditions, as upstream wind turbines could be turned off, could be

yawing or could be pitching. This further means that they show different correlation curves and should be sorted into different correlation states. Thus, to group data according to the underlying flow conditions, which define the different correlation states, the introduced parameters (standard deviation of the power output fluctuations of wind turbines in a pair and the normalised power difference of wind turbines in a pair) were combined with the clustering algorithm k-means. The clustering showed similar results for the wind directions 90° and 270°. The clusters had distinguishable values in the standard deviation of the power output fluctuations and in the normalised power output differences of the wind turbines, which were directly related to the average correlation curve per cluster. Increased standard deviations of the power output fluctuations combined with the small normalised power difference of the wind turbines in a pair showed the most defined correlations with the highest peak (Cluster 1). This combination was found for wind turbine pairs located further downstream in the wind farm and some wind turbine pairs from rows towards the front. For 90°, the peak of the correlation increased via clustering from 0.16 to 0.32, and for 270° the peak of the correlation increased from 0.21 to 0.41. A value of 0.41 is close to the correlations found in the LES study by Lukassen et al. (2018) and experiments by Bossuyt et al. (2017b), which were between 0.5 and 0.55 for similar wind turbine spacing and similar wind speeds. In addition, for both wind directions, a cluster of non-correlated wind turbines (Cluster 5) was found, which mainly consists of wind turbine pairs in the first rows of the wind farm. Clusters 2, 3 and 4 were not as significant as Clusters 1 and 5 and showed distinguishable correlation curves with their peaks ranging from 0.14 to 0.22 for 90° and from 0.2 to 0.31 for 270°.

Hence, we found that to analyse correlation states of power output fluctuations of streamwise wind turbine pairs in varying flow conditions, the standard deviation of the power output fluctuations of wind turbines in a pair as well as the normalised power difference of the wind turbines in a pair have been proven to be suitable to identify correlation states. Furthermore, the data-driven $k$-means clustering approach enables an automated grouping of the data into correlation states based on these parameters. As an outlook, further analysis of the space-time correlations within an offshore wind farm could help control wind turbines, e.g. for power output fluctuation management or active wake control. Also, knowledge about the correlation of wind turbine pairs allows short-term power output fluctuation forecasting within the wind farm and interactive wind turbine control. The presented findings can be enhanced in the future by adding Lidar or radar measurements to access independent wind direction and wind speed measurements. Moreover, the analysis of correlation states might be extended to include the correlation of wind turbine pairs with multiple inter-turbine distances and the correlation of non-aligned wind turbine pairs. Clustering of correlation states can be further investigated by increasing the number of clusters to $k > 5$. Results for $k = 6$ indicated that the statistics of the upstream and downstream wind turbine of a pair have a different influence on its correlation. In addition, it is worth considering alternative clustering methods like $k$-medoids (Kaufman and Rousseeuw, 2008), which is less sensitive to outliers or Density-Based Spatial Clustering of Applications with Noise (DBSCAN) (Ester et al., 1996) which is also less sensitive to outliers and has no fixed cluster shapes and fixed number of clusters. Using these algorithms could improve the clustering of the intervals and more defined correlation curves or could identify further clusters. Furthermore, measurements on the boundary layer conditions help assess the influence of wind turbine wakes on the space-time correlations of power output fluctuations with the additional knowledge on the atmospheric stability.

## Appendix A:  Wind turbine pairs

To calculate the power output fluctuation correlation, wind turbine pairs are chosen according to the respective wind direction. In total, 66 wind turbine streamwise pairs can be defined. Table A1 depicts the definition of wind turbine pairs for wind direction 270°. For wind direction 90°, the same pairs are chosen but with switched wind turbine order. E.g. for pair 1 for 270°, wind turbine 1 is the upstream wind turbine and wind turbine 2 is the downstream wind turbine. For 90°, wind turbine 2 is the upstream wind turbine and turbine 1 is the downstream wind turbine.

**Table A1.** Definition of streamwise wind turbine pairs for wind direction 270°.

| Pair | 01 | 02 | 03 | 04 | 05 | 06 | 07 | 08 | 09 | 10 | 11 | 12 | 13 | 14 |
|------|------|------|------|------|------|------|------|------|------|------|------|------|------|------|
| WTs | 01, 02 | 02, 03 | 03, 04 | 04, 05 | 05, 06 | 06, 07 | 07, 08 | 09, 10 | 10, 11 | 11, 12 | 12, 13 | 13, 14 | 14, 15 | 16, 17 |
| Pair | 15 | 16 | 17 | 18 | 19 | 20 | 21 | 22 | 23 | 24 | 25 | 26 | 27 | 28 |
| WTs | 17, 18 | 18, 19 | 19, 20 | 20, 21 | 21, 22 | 23, 24 | 24, 25 | 25, 26 | 26, 27 | 27, 28 | 28, 29 | 30, 31 | 31, 32 | 32, 33 |
| ´Pair | 29 | 30 | 31 | 32 | 33 | 34 | 35 | 36 | 37 | 38 | 39 | 40 | 41 | 42 |
| WTs | 33, 34 | 34, 35 | 35, 36 | 37, 38 | 38, 39 | 39, 40 | 40, 41 | 41, 42 | 42, 43 | 44, 45 | 45, 46 | 46, 47 | 47, 48 | 48, 49 |
| Pair | 43 | 44 | 45 | 46 | 47 | 48 | 49 | 50 | 51 | 52 | 53 | 54 | 55 | 56 |
| WTs | 49, 50 | 51, 52 | 52, 53 | 53, 54 | 54, 55 | 55, 56 | 56, 57 | 58, 59 | 59, 60 | 60, 61 | 61, 62 | 62, 63 | 64, 65 | 65, 66 |
| Pair | 57 | 58 | 59 | 60 | 61 | 62 | 63 | 64 | 65 | 66 | | | | |
| WTs | 66, 67 | 67, 68 | 69, 70 | 70, 71 | 71, 72 | 73, 74 | 74, 75 | 76, 77 | 77, 78 | 79, 80 | | | | |

## Appendix B:  Statistical dependence of 600 s intervals

The analysed 600 s intervals are not statistically independent as they overlap by 599 s in the extreme case. Also, thinking of bigger gusts evolving through the wind farm, it is most likely that wind turbine pairs experience similar correlation states when being affected by the gust. To clarify the influence of the overlapping of the considered intervals, we performed the calculations again using only non-overlapping intervals. The following figure (Fig. B1) compares the results for non-overlapping and overlapping intervals exemplary for wind direction 270°. Figure B1a displays the comparison of the average correlation curves per wind farm row. Figure B1b displays the comparison of the average correlation curves per cluster. In general, the results of the non-overlapping intervals are similar to the results of the overlapping intervals and differ at most by about 10%. However, Fig. B1a shows that this data set is at the limit of representing the correlations as the average correlation curves start to wiggle for $\tau_{norm} > 2$ due to the low number of data points. In total, only 8121 non-overlapping 600 s intervals are available for 270°, which resemble a measurement time of about 56 days. For all wind turbine pairs, 11514 intervals are available as multiple wind turbine pairs are available in the same interval.

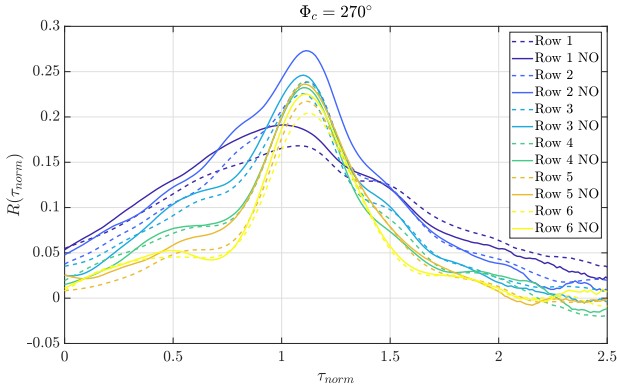
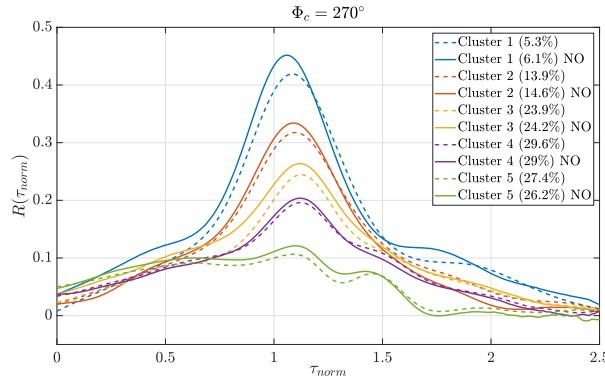

(a) Average power output fluctuation correlation per wind farm row.

(b) Average power output fluctuation correlation per cluster.

**Figure B1.** Comparison of the average power output fluctuation correlation for wind direction interval 270° of non-overlapping and overlapping intervals. For both plots, the average correlation curves for non-overlapping intervals are marked with 'NO' and plotted with a dashed line. The average correlation curves for overlapping intervals are plotted in both cases as a solid line.

## Appendix C: Effect of the numbers of clusters

As mentioned in Sect. 4, the number of clusters chosen for the present analysis was $k = 5$. This decision was made based on the results for $k = 6$ presented in Figure C1 and Fig. C2. For wind direction 90°, six separable correlation curves are found. Comparing Fig. C1 to Fig. 7, it shows that Cluster 2 of Fig. 7 seems to be separated into two clusters (Cluster 2 and 3 of Fig. C1).

For wind direction 270°, only five clearly separable correlation curves are found where one is overlapped by a very similar one. Comparing Fig. C2 to Fig. 8, it shows that Cluster 3 of Fig. 7 seems to be separated into two similar clusters (Cluster 3 and 4 of Fig. C2). The new clusters also do not reveal any further characteristics.

Looking at the statistics of the correlation curves listed in Tab. C1, it can be found that for wind direction 90°, Cluster 2 shows a higher standard deviation of the power output fluctuations for wind turbine A instead of B, while Cluster 3 shows a higher standard deviation of the power output fluctuations for wind turbine B instead of A, similar to all other clusters. For the wind direction 270°, Cluster 4 shows a higher standard deviation of power output fluctuations for wind turbine A instead of B, while Cluster 3 shows a higher standard deviation of power output fluctuations for wind turbine B instead of A, similar to all other clusters.

The correlation curves and statistics imply that the further separation of the statistics with $k > 5$ does not reveal any correlation states that are more significant than those found for $k = 5$. Clustering with $k > 5$ might further distinguish the flow states for wind turbine pairs based on the standard deviations of power output fluctuations of wind turbines A and B. However, this is not further investigated, as this effect is not included in the scope of the work presented here.

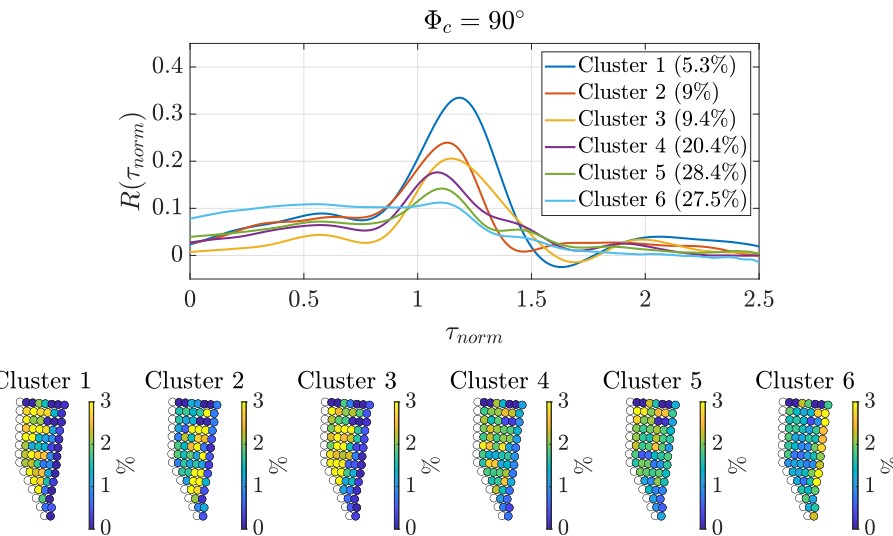

**Figure C1.** Clustering for wind direction interval around 90° with randomly sorted parameters and $k = 6$. $\Phi_c$ depicts the centre of the wind direction interval.

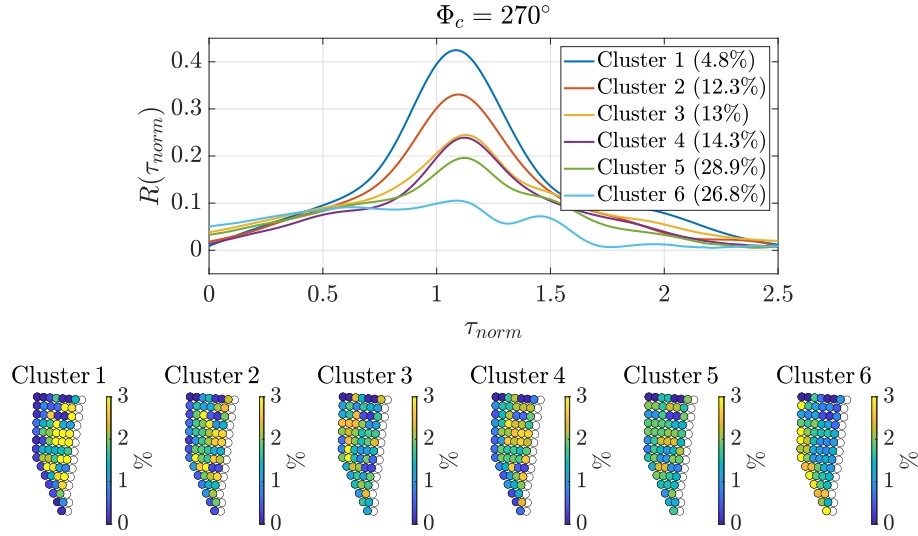

**Figure C2.** Clustering for wind direction interval around 270° with randomly sorted parameters and $k = 6$. $\Phi_c$ depicts the centre of the wind direction interval.

**Table C1.** Averaged wind turbine statistics computed for wind direction intervals around 90° and 270° and $k = 6$, with $A$ as the upstream wind turbine and $B$ as the downstream wind turbine. $\sqrt{\langle P'^2_A \rangle_{\Delta t_{600}}}$ is the standard deviation of the power output fluctuations of wind turbine $A$ over a 600 s interval $\Delta t_{600}$ (analogue for wind turbine $B$ for the same 600 s intervals, respectively). $\langle P_A \rangle_{\Delta t_{600}}$ and $\langle P_B \rangle_{\Delta t_{600}}$ are the average power of wind turbines $A$ and $B$ over the same 600 s intervals. $\langle ... \rangle_{cluster}$ denotes the average of the statistics over all available time intervals of the wind turbine pairs. Note that here 90° and 270° again refer to 20° wind direction intervals from 80° to 100° and from 260° to 280°.

| Cluster | $\left\langle \sqrt{\langle P'^2_A \rangle_{\Delta t_{600}}} \right\rangle_{cluster}$ [kW] | | $\left\langle \sqrt{\langle P'^2_B \rangle_{\Delta t_{600}}} \right\rangle_{cluster}$ [kW] | | $\left\langle \frac{\langle P_A \rangle_{\Delta t_{600}} - \langle P_B \rangle_{\Delta t_{600}}}{\langle P_A \rangle_{\Delta t_{600}}} \right\rangle_{cluster}$ | |
|---|---|---|---|---|---|---|
| | 90° | 270° | 90° | 270° | 90° | 270° |
| 1 | 523 | 541 | 526 | 547 | 0.02 | 0.04 |
| 2 | 327 | 394 | 435 | 412 | 0.02 | 0.06 |
| 3 | 393 | 247 | 324 | 343 | 0.09 | 0.06 |
| 4 | 240 | 317 | 263 | 263 | 0.11 | 0.07 |
| 5 | 152 | 194 | 168 | 211 | 0.13 | 0.07 |
| 6 | 72 | 116 | 84 | 132 | 0.14 | 0.11 |

*Author contributions.* JKS developed the underlying method, performed the data analyses and wrote the paper. LJL provided intensive consultation on the development of the method and the scientific analyses. MKr and MKü provided intensive reviews on the scientific analyses. LJL had a supervising function.

*Competing interests.* The authors declare that they have no conflict of interest.

*Acknowledgements.* We performed parts of the work within the research project 'OWP Control' (FKZ 0324131A) funded by the German Ministry of Economic Affairs and Energy basis of a decision by the German Bundestag. Parts of the computations were performed on the high-performance computing system EDDY of the University of Oldenburg which is part of the project 'WIMS-Cluster' (FKZ 0324005) founded by the Federal Ministry of Economic Affairs and Energy. We acknowledge the wind farm operator Global Tech I Offshore Wind GmbH for providing SCADA data and their support of the work.

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
