# Peer review of "Correlations of power output fluctuations in an offshore wind farm using high-resolution SCADA data"

_Wind Energy Science, 2020_

## Referee Comment (RC1) · Anonymous Referee #1 · 15 Nov 2020

In the manuscript the authors analyze the correlations of power output fluctuations in the offshore wind farm Global Tech I wind farm based on SCADA data obtained over an eight month period. The required data analysis to study these phenomena in SCADA data is very challenging. The topic is relevant and interesting for the wind energy community and suitable for publication in Wind Energy Science Discussions.

That being said, I believe the authors should carefully revise the manuscript before it can be considered for publication. The current manuscript lacks clarity on various important points (see below), which need clarification and more detailed discussion.

Please find below the corresponding list of comments, recommendations , suggestions,

[Figure]

and questions.

* In the abstract it is unclear what "clustering algorithm k-means" is. This method may not be known to all readers and requires explanation. * Even after reading the entire manuscript the goal and exact outcome of this analysis remains vague. It seems that during this analysis the data is sorted, based on the power fluctuations, and that based on this division the power correlations in certain parts of the data are more pronounced than in other parts of the data set. What does this tell us exactly? I guess from figure 8 and 9 we can see that power correlations are more pronounced for certain turbines than for others (however, data from the same turbines seems to be included in different clusters). So in figure 8 cluster 5 is dominated by turbines on the first row, but a similar observation can be made from figure 6a already. So it is not quite clear what the added value of the "k-means clustering algorithm" analysis in this work is. This should be clarified, or that analysis should be removed from the manuscript.

Line 15: "7 wind farms were connected to the grid"? ==> This seems low. I guess only wind farms above a certain size are included. Line 47: It seems a bit strange that wind direction changes have only a a little effect on power output fluctuations. In fact, I would say that the results presented in the current manuscript indicate the opposite. What kind of wind farm did Dai et al. (2017) considered?

Line 76: It is stated that U is not measured, but calculated from the measured power. Please clarify how exactly that is done, and to what degree this procedure could affect the presented findings. Line 78: "However, it can still be used for assessing the effect of the wind speed on the wind turbine." ==> what effect of the wind speed on the wind turbine are you referring to (you use the power to get the wind speed).

Figure 1: Throughout the manuscript the authors focus on the 90 degree and 270 degree wind directions. Figure 1 suggests the wind farm is not perfectly aligned with the 270 / 90 degree direction (for example turbine 15 seems a little lower than turbine 9). Is this indeed the case, and if so, why did the authors not select the wind directions

corresponding to the wind farm alignment in the red box of figure 1.

Table 1: Is the filtering really performed with "no yawing", or is this also practically implemented with some low threshold?

Line 108-111: Please clarify what you mean? Do you refer to turbines which are limited in production because of other consideration than their own individual controller?

Line 163-165: in the explanation of t_norm you refer to some time averages. What time averages do you exactly use (over the 600 time second window)?

Line 171: Why is this reference speed chosen and not a higher value? In figure 4 we see the peak values are observed at t_norm=1.1 to 1.2. Can you give some more detailed consideration as what we can infer from this value.

Figure 2 (and other places): a total of over 9 million intervals is mentioned. Are these statistically independent, or not? Please clarify.

Figure 4: Why are the results not symmetric? I.e. for example 260 shows pronounced peak, but 280 degree does not. This aspect should be discussed. See also my above question on the selection of the 90 and 270 degrees wind directions.

Line 203: For wind directions approaching $0°$ and $180°$ the wind turbines in a pair are oriented more perpendicular to the wind. Fluctuations reach the downstream turbine earlier. ==> If the turbines are perpendicular for these wind directions, what is then the "downstream" turbine?

Line 224: Normalized by what?

Table 2 and other tables: If possible, I believe it would also be useful to mention the average power outputs of the wind turbines A and B

Line 250: "Even though such wind turbines are filtered out for the analysis, they still influence the surrounding wind turbines in an unpredictable way." ==> Please clarify the meaning of this statement. If the data is filtered such that all turbines are operations

what effects are then not filtered for?

Figure 7 (line 260-263): If figure 7 is just showing two lines already shown in figure 4 why do we need this additional figure?

Line 264-265: "into a reasonable set of groups and a greater number of cluster did not lead to further clusters of importance for the present analysis (see appendix B)." ==> Please clarify this statement. Also after reading the appendix this was not quite clear to me. What does "reasonable" mean? And what is a cluster of importance?

Line 308: Please be specific so the conclusion section can be read independently.

Line 320: It is unclear to me what can be learned from these different clustering approaches. Why are these specific ones suggested?

---

## Referee Comment (RC2) · Mark Kelly (Referee) · 30 Nov 2020

**General comments**

The analysis of power output correlations across a wind farm can certainly be relevant within wind energy.
This is certainly a challenging task given the data analyzed and its limitations; the latter has been only partly addressed.

The abstract does not appear to include motivation for the study, and the introduction also lacks clear motivation and/or justification; why/how is this work relevant?
Unfortunately there are a number of issues with the submitted draft, such that it requires at least major revisions. A number of plots showing pair-wise power output correlation versus normalized lag are shown; but their statistical significance is not evident, nor argued thoroughly, nor put within any context of the scales of atmospheric inflow fluctuations. (One can see by the repeated trends, of course, that the plotted $R(\tau_{norm})$ are not simply noise, despite being at most $\sim$0.1–0.4 depending on the data selection.) The methods used are not explained in sufficient detail, with references to such also lacking. As currently reported, the study would not be reproducible by a reader. The results need to be more clearly presented, and also interpreted, by the authors—in addition to the inclusion of relevant details, motivation, and significance of the study.

There are errors in language throughout; I have included some correction examples in the last section below, but suggest that the next version be proofread by somebody with near-native level fluency.

**Specific comments**

l.8: what is meant by 'correlation states'? Lines 7–9, starting with 'Most importantly', should be reworded. To be more direct: you are using a clustering algorithm (k-means clustering) to group similarly correlated turbine pairs, in order to examine the spatial variation of correlations between turbines and determine the key parameters affecting such correlations.

l.9–10: The phrase 'next to' isn't appropriate here; it seems you're wanting to say 'in addition to' or similar. Also, do you mean the location of a turbine pair is most important, or the relative locations/distance?

l.38: in terms of correlations, the wind farm was not "infinitely large", was it? I.e., were there not periodic boundary conditions used in the LES?

l.40: by 'variance of the wind velocity', do you not mean strength of turbulence in the

prescribed inflow?

l.55,l.64 and elsewhere: you haven't (yet) defined "correlation state"

l.71: why "non-axisymmetric"? Do you mean asymmetric?

l.74-76 and afterward: How did you calculate the wind direction and U? What transfer function(s) were used, and how was this blended with speed implied by measured power?

l.77: do you mean that the yaw error was not used in the nacelle wind speed transfer function/correction?

l.87–89: did you consider the across-farm variation in wind direction, with suitable averages of upwind turbines?

l.111 and Table 1: how did you determine the pitch threshold?

l.112–116 and Table 1: what is the yaw misalignment threshold? What about $\varphi$ change per 600s?

l.120–121: why not report the variation of angles, instead of only the mean with 10-degree tolerance?

l.155–159: mention the use of Taylor's hypothesis here—and the assumption you're thus using over the entire range of $\tau_{norm}$. There are several references you should check (and cite) regarding this.

l.163–166: Your statement about $\tau_{norm} > 1$ has an implication: if you used the average speed between turbines A and B (instead of $U_B$), then wouldn't $\tau_{norm}$=1, unless the propagation speed is somehow otherwise affected? It's difficult to defend using just $U_B$.

l.181: why 20deg interval? How does this compare to the variation of $\varphi$ across all turbines for a 10-minute periods? The latter is likely important to describe the inflow

state and variability of the wind field (not to mention yaw error).

p.8/l.201–203: Indeed $\rho$=0.2 is not generally considered to be statistically significant. Have you tried windows of different lengths (other than 300s)? Have you considered the integral timescale of the incoming turbulence? What about the wake turbulence?

Table 3: is it reasonable to include 3 significant digits in the correlations listed? (or 5 in the power?) Such second-order statistics do not converge so easily within 10 minutes...

l.253–255: While this reviewer has some familiarity with k-means clustering, how is a reader supposed to be able to understand (let alone repeat) the analysis reported here? Please include appropriate references and details.

l.293 and elsewhere: instead of just 'filtering', perhaps you should use a term like 'data selection process'; recall that in the spectral sense a *filter* means something else, and such filtering could be expected for the kind of analysis you do here.

l.296: "Deviations from the 90° and 270° wind directions result in a decreased correlation" is out of context here; this and the statement after it do not make sense as written.

l.299–300: pairs within the first (upwind) row of turbines have little correlation not simply because of the 'free-stream' inflow, but because their spacing is greater than the transverse integral length scale of the turbulence.

l.308–309: simply using 'previously defined' or 'chosen statistics' is somewhat obfuscatory and not really appropriate in a conclusion/summary; it does not explain to the reader which 'statistics' you are considering.

l.309–311: why not just use joint distributions, and conditional statistics? How is k-means more helpful?

l.311–312: this statement is not understandable without more context; please help the reader, and interpret it also.

l.320–323: why were no details about this given in section 4?

**Technical corrections**

l.6: 'correlation is' should be 'power correlations are'; 'towards' is not appropriate

l.15: second 'and' should be 'while' or similar.

l.23: 'the respective' should be 'a respective'

l.27–29: need to re-word run-on sentence, including e.g. appropriate commas

l.33: remove 'a' before 'high'

l.46: 'of' should be 'between'

l.81: reference missing year

l.400: incorrect journal and doi for Valldecabres reference

l.403–404: the WindEurope reports appear to be missing some identifying information (e.g. doi, report number, etc.).

---

## Author Comment (AC1) · 23 Jan 2021

**Correlations of power output fluctuations in an offshore wind farm using high-resolution SCADA data**

Authors: Janna K. Seifert, Martin Kraft, Martin Kühn, and Laura J. Lukassen

DOI: 10.5194/wes-2020-96
* * *
**Authors response to referee comments**

Dear referees, we appreciate the constructive comments which help to improve our paper. Below we will discuss all comments in detail.
* * *
**Authors response to comments from referee #1**

**RC1** *\* In the abstract it is unclear what "clustering algorithm k-means" is. This method may not be known to all readers and requires explanation. \* Even after reading the entire manuscript the goal and exact outcome of this analysis remains vague. It seems that during this analysis the data is sorted, based on the power fluctuations, and that based on this division the power correlations in certain parts of the data are more pronounced than in other parts of the data set. What does this tell us exactly? I guess from figure 8 and 9 we can see that power correlations are more pronounced for certain turbines than for others (however, data from the same turbines seems to be included in different clusters). So in figure 8 cluster 5 is dominated by turbines on the first row, but a similar observation can be made from figure 6a already. So it is not quite clear what the added value of the "For wind directions approaching 0° and 180° the wind turbines in a pair are oriented more perpendicular to the wind and fluctuations reach both wind turbines A and B at nearly the same time.-means clustering algorithm" analysis in this work is. This should be clarified, or that analysis should be removed from the manuscript.*

    **AC:** Thank you for pointing this out. Fig. 6 shows that the position of the wind turbine pairs has an influence on the correlation of their power output fluctuations and that the correlation curves are more pronounced for wind turbine pairs at the back of the wind farm. However, the presented curves are averages of several correlation states which depend on the flow conditions the wind turbine pairs are exposed to. Combining the introduced parameters: the standard deviation of the power output fluctuations and the power difference of the wind turbines in a pair, with the clustering algorithm For wind directions approaching 0° and 180° the wind turbines in a pair are oriented more perpendicular to the wind and fluctuations reach both wind turbines A and B at nearly the same time.-means, the data can be separated according to the underlying correlation states, independently from the position of the wind turbines. The clusters with the highest correlations mostly consist of wind turbine pairs at the back of the wind farm. Their correlation is significantly increased compared to the average power output fluctuation correlation of all pairs in the last row of the wind farm. We revised the abstract and introduction to clarify the added value of the *k*-means clustering algorithm:

> p.1, ll.1f: Space-time correlations of wind turbine pairs provide information on the flow conditions within a wind farm and on the interactions of the wind turbines. Such information plays an important role for the control of wind turbines and short-term load or power forecasting. However, the challenge to analyse space-time correlations of power output fluctuations of wind turbine pairs in a free field wind farm are the highly varying flow conditions. Here, we present an approach to investigate space-time correlations of power output fluctuations of wind turbine pairs in free field based on high-resolution SCADA data, which overcomes the challenge of highly variable flow conditions within the wind farm. Using eight months measurements from an offshore wind farm with 80 wind

turbines, the influences of different parameters on the correlation of power output fluctuations are analysed. Wind direction investigations show that correlations of power output fluctuations of wind turbine pairs are highest for streamwise aligned pairs and decrease towards spanwise pairs. Further, it is found that the correlation of power output fluctuations of streamwise aligned wind turbine pairs depends on the location of the wind turbines within the wind farm as well as the inflow conditions (free-stream or wake). The main outcome is that the correlation of streamwise aligned wind turbine pairs can be characterised by the standard deviations of the power output fluctuations and the power difference of the wind turbines in a pair. Evaluating these parameters with the data-driven clustering algorithm *k*-means, wind turbine pairs with similar power output fluctuation correlations are grouped depending on these parameters and independent from their location. These groups are here referred to as correlation states. With this approach we account for the highly variable flow conditions inside a wind farm which influence the correlations in an unpredictable way. As a final result we shows that these parameters lead to clearly distinguishable correlation states.

p.2, ll. 56f: In our work, we analyse 1 Hz wind farm SCADA data to describe space-time correlations of power output fluctuations of wind turbine pairs. In contrast to the wind tunnel measurements by Bossuyt et al. (2017a) and the LES analysis by Lukassen et al. (2018) mentioned above, the data set processed here includes unstable inflow conditions, dynamically operating wind turbines as well as changing flow conditions within the wind farm. Furthermore, there may be potential measurement inaccuracies. The result is a large and highly complex data set. In this paper we investigate the influencing factors on the correlation of power output fluctuations of wind turbine pairs and introduce parameters to distinguish different correlation curves, here called correlation states.

**RC2** *Line 15: "7 wind farms were connected to the grid"? ==> This seems low. I guess only wind farms above a certain size are included.*

**AC** According to WindEurope, 7 wind farms were fully grid-connected and additional 3 have been partially connected to the grid. Furthermore, the construction of 5 other wind farms started (Ramírez et al., 2020). For clarification we added the information that this number refers only to grid-connected wind farms:

p.1 ll. 20f: Considering offshore wind power in 2019, the capacity in Europe has increased by 3.627 GW, and a total of 7 wind farms were fully connected to the grid, while the average size of wind farms increased to 621 MW (Ramírez et al., 2020).

**RC3** *Line 47: It seems a bit strange that wind direction changes have only a little effect on power output fluctuations. In fact, I would say that the results presented in the current manuscript indicate the opposite. What kind of wind farm did Dai et al. (2017) considered?*

**AC** Dai et al. (2017) investigated the influence of wind direction fluctuations around a mean wind direction on the power output fluctuations of single wind turbines. Whereas we investigate the influence of different wind directions on the correlation of wind turbine pairs. We analyse wind different wind direction intervals with a size of 20° and show that different wind directions have an influence on the correlation of the wind turbine pairs. For wind directions around 90° and 270°, the wind turbines of the considered pairs are streamwise aligned to the wind direction and show a high correlation. Whereas for 0° and 180° the wind turbines of a pair are parallel to the wind direction and show nearly no correlation.

Thus, it is correct that the wind direction has an influence on the correlation of the considered wind turbine pairs, but the influence of wind direction fluctuations was not further investigated here. To clarify this, we changed the following texts:

> p. 2 , ll. 51f: Dai et al. (2017) analysed 1 Hz wind farm SCADA data with respect to the influence of wind speed fluctuations and wind direction fluctuations on wind turbine power output fluctuations of single wind turbines. They showed a direct relation between wind speed fluctuations and power output fluctuations in the partial load regime.

> p. 3, ll. 92f: The nacelle based wind direction $\varphi$ is estimated based on the measurements of two 2D sonic anemometers installed behind the rotor of each wind turbine. These measurements have to be treated with care as the measured flow behind the rotor is disturbed by the rotation of the rotor and the nacelle itself. Thus, it is only an estimation of the wind direction and yaw of the wind turbine. However, as shown by Dai et al. (2017), wind direction fluctuations at reasonable yaw angles ($< 45°$) have only little effect on the power output fluctuations of wind turbines and thus inaccuracies in $\varphi$ have no major influence on the performed analysis. The combined measurements of $\theta_i$ and $\varphi_i$ define the wind direction $\Phi_i$ at the $i$-th wind turbine.

**RC4** *Line 76: It is stated that U is not measured, but calculated from the measured power. Please clarify how exactly that is done, and to what degree this procedure could affect the presented findings.*

    **AC** Unfortunately, $U$ is provided as a variable within the data set. It is not measured but directly related to the measured power and reconstructed from the wind turbine (controller) settings. Details on the reconstruction are not available. Depending on $U$, the correlation curves stretch or shrink (see p.6, eq. 3). Thus, if the recalculated wind speed differs from the actual wind speed, the normalised correlation curve might be slightly shifted to a larger $\tau_{norm}$. However, $U$ affects the normalisation of the correlation curves in a consistent manner for all wind turbines. For this reason, we consider $U$ as reasonable variable in the context of this analysis. We revised the description of $U$ and added how it affects the normalisation of correlation curves:

> p. 3, ll. 81f: The processed signals include the generated power $P$, the azimuth angle of the wind turbines (i.e. the nacelle direction) $\theta$, the nacelle based wind direction $\varphi$ (measured relative to $\theta$), the pitch angle $\beta$ of each blade, and a reconstructed wind speed $U$.
> The reconstructed wind speed $U$ is not directly measured but provided as a variable which results from the measured power and control variables of the wind turbine. Due to that, $U$ is considered as an approximated and idealised value which does not include wind speed independent power reduction, e.g. by misalignment of the wind turbine due to measurement errors of the wind direction. In the context of this work, it can still be used for assessing the effect of the wind speed on the correlations of power output fluctuations of wind turbine pairs which is further discussed in Sect. 2.2.

> p.7, ll. 179f: As mentioned before, $U_B$ is reconstructed and might differ from the actual wind speed affecting the wind turbines. However, in the context of this normalisation the effect on the resulting correlations curves is marginal as the correlation curves may only be slightly shifted due to the deviation to the real wind speed.

**RC5** *Line 78: "However, it can still be used for assessing the effect of the wind speed on the wind turbine." ==> what effect of the wind speed on the wind turbine are you referring to (you use the power to get the wind speed).*

    **AC** Thank you for this remark. It should have said the influence of the wind speed on the correlation of power output fluctuations of wind turbine pairs. We corrected this sentence in the context of **RC4** as follows:

> p.3, ll. 87f: In the context of this work, it can still be used for assessing the effect of the wind speed on the correlations of power output fluctuations of wind turbine pairs which is further discussed in Sect. 2.2.

**RC6** *Figure 1: Throughout the manuscript the authors focus on the 90 degree and 270degree wind directions. Figure 1 suggests the wind farm is not perfectly aligned with the 270 / 90 degree direction (for example turbine 15 seems a little lower than turbine9). Is this indeed the case, and if so, why did the authors not select the wind directions corresponding to the wind farm alignment in the red box of figure 1.*

**AC** It is correct, that wind turbine 15 and wind turbine 9 are not exactly horizontally aligned. However, we chose to stick wind wind directions 90° and 270° since the best correlation of power output fluctuations were found for these directions as shown in Fig. 3. Also, as we consider wind direction intervals of 20° around 90° and 270°, the 'ideal' wind direction for the slightly horizontal misaligned wind turbines is included within the intervals around 90° and 270°.

**RC7** *Table 1: Is the filtering really performed with "no yawing", or is this also practically implemented with some low threshold?*

**AC** Yes, this filtering is performed with "no yawing" within the considered correlation intervals. This means both wind turbines were not allowed to yaw. To clarify this, we changed the table entry to:

> p. 5, Tab. 1:
>
> | Signal | Power | Pitch | Yaw |
> |---|---|---|---|
> | Settings | $0.5\,\text{MW} \leq P \leq 4.5\,\text{MW}$ | $\beta < -1.3°$ | $\theta = const.$ |

**RC8** *Line 108-111: Please clarify what you mean? Do you refer to turbines which are limited in production because of other consideration than their own individual controller?*

**AC** With derated wind turbines we refer to wind turbines whose controller settings have been manually changed so that the maximum power is limited to a certain value lower than the nominal power. This means derated wind turbines start pitching earlier than non-manipulated wind turbines as their maximum power limit is reached at lower wind speeds. To clarify this, we revised the text as follows:

> p. 5, ll. 119f: The previously defined, limited power range still includes derated wind turbines. For derating wind turbines, their controller is manually changed so that their maximum power is limited to a certain value lower than their nominal power. Due to that, wind turbines might start pitching already in the previously defined load range as their newly set power limit is reached already at lower wind speeds. Hence, to fully exclude pitching wind turbines, the data is filtered for any pitching activity. Please note that for this specific data set this implies that $\beta < -1.3°$.

**RC9** *Line 163-165: in the explanation of $\tau_{norm}$ you refer to some time averages. What time averages do you exactly use (over the 600 time second window)?*

**AC** When calculating $\tau_{norm}$ for a certain correlation time interval of 600 s starting at time $t$, we average the reconstructed wind speed at wind turbine B over 300 s for $t$ in the discretised interval $[t_j, t_j + 299\,\text{s}]$ with $\langle U_B(t+\tau) \rangle_{\Delta t_{300}}$, the discretised interval of $U_B(t+\tau)$ e.g. for $\tau = 100$ is $[t_j + 100, t_j + 100 + 299\,\text{s}]$. We corrected eq. 3, removed eq. 4 and revised the paragraph as follows to clarify the procedure:

p. 6, ll. 166f: Similar to Taylor's hypothesis (Taylor, 1938) we assume that depending on the wind speed, wind structures responsible for power output fluctuations measured at an upstream wind turbine A, take some time to travel the distance to the downstream wind turbine B. Hence, to compare correlations at different wind speeds and different wind turbine distances, the time lag $\tau$ is normalised for each time interval starting at $t_j$ individually

$$\tau_{norm,intv} = \tau \cdot \frac{\langle U_B(t+\tau) \rangle_{\Delta t_{300}}}{x_{AB}} \qquad (3)$$

where $\tau_{norm,intv}$ is the normalized time lag, $\langle U_B(t+\tau) \rangle_{\Delta t_{300}}$ is the average reconstructed wind speed from a certain (downstream) wind turbine B for a time interval $\Delta t_{300} = 300$ s for $t$ in the discretised interval mbox$[t_j, t_j + 299$ s$]$ and a certain lag $\tau$. This means for a certain $\tau$, the averaging interval of $\langle U_B(t+\tau) \rangle_{\Delta t_{300}}$ is $[t_j + \tau, t_j + \tau + 299$ s$]$. $x_{AB}$ is the distance between wind turbine A and wind turbine B.

Due to this definition of $\tau_{norm,intv}$ and $\tau_{norm}$ (see Eq. 4), the peak of the correlation curves is expected to be found at $\tau_{norm} = 1$ if the advection speed of the wind speed fluctuations matches the wind speed affecting B. Thus, in partial load situations where wind turbine B is in the wake of wind turbine A, the peak is expected to be at $\tau_{norm} > 1$. Here, the reduced wind speed in the wake recovers slowly, so that the wind speed affecting wind turbine B, i.e., $U_B$ is already partly recovered and hence larger than the advection speed of the fluctuations. As mentioned before, $U_B$ is reconstructed and might differ from the actual wind speed affecting the wind turbines. However, in the context of this normalisation the effect on the resulting correlations curves is marginal as the correlation curves may only be slightly shifted due to the deviation to the real wind speed.

In a next step, the correlation curves with the normalised lag $\tau_{norm,intv}$ are discretised using a histogram with a reference time lag of

$$\tau_{norm} = \tau \cdot \frac{U_{max}}{x_{AB,mean}} \qquad (4)$$

where $\tau$ is the time lag (0 s to 300 s), $U_{max}$ is an artificially introduced velocity which has to be at least equal to the maximum possible wind speed to fit all normalised curves (here $U_{max} = 13$ ms$^{-1}$). $x_{AB,mean}$ is the average distance between wind turbine A and wind turbine B of the considered wind turbine pairs. Note that $\tau_{norm,intv}$ is only used for stretching and shrinking of the correlation curves and that $\tau_{norm}$ is used only for binning of the stretched or shrunk correlations.
* * *
p. 7, ll. 195f: In the following, we average correlations over a wind direction interval of 20° and all available time intervals of the considered wind turbines (either all wind turbines or a selection of wind turbines). The averaged correlation is noted as $R(\tau_{norm})$.

**RC10** *Line 171: Why is this reference speed chosen and not a higher value? In figure 4 we see the peak values are observed at $t_{norm}$=1.1 to 1.2. Can you give some more detailed consideration as what we can infer from this value.*

    **AC** As described in Eq. 4 in the revised Sect. 2.2 (see **RC9**), the peak value of the correlation curves is dependent on the relation of the reconstructed wind speed at wind turbine B ($U_B$) and the advection speed of fluctuations between wind turbine A and B. If the wind speed at wind turbine B is overestimated the peak is found at $\tau_{norm} > 1$. The curves in Fig. 4, with peaks at $\tau_{norm} = 1.1$ and $\tau_{norm} = 1.2$ indicate that $U_B$ is as expected overestimating the advection speed of wind speed fluctuations. Note that $\tau_{norm,intv}$ is used for the stretching and shrinking whereas $\tau_{norm}$ is only used to bin the stretchted or shrunk curves. We added for clarification:

p. 7, ll. 187f: Note that $\tau_{norm,intv}$ is only used for stretching and shrinking of the correlation curves and that $\tau_{norm}$ is used only for binning of the stretched or shrunk correlations.

**RC11** *Figure 2 (and other places): a total of over 9 million intervals is mentioned. Are these statistically independent, or not? Please clarify.*

    **AC** The analysed 600 s intervals are not statistically independent as they overlap by 599 s in the extreme case. Also, thinking of bigger gusts evolving through the wind farm, it is most likely that wind turbine pairs experience similar correlation states when being affected by the gust. To clarify the influence of the overlapping of the considered intervals we performed the calculations again using only non-overlapping intervals. The following figure (Fig. 1) shows a comparison of the results for non-overlapping and overlapping intervals exemplary for wind direction 270°. Figure 1a displays the comparison of the average correlation curves per wind farm row. Figure 1b displays the comparison of the average correlation curves per cluster. In general, the results of the non-overlapping intervals are similar to the results of the overlapping intervals and differ at most by about 10%. However, the comparison in Fig. 1a shows that this data set is at the limit of representing the correlations as the average correlation curves start to wiggle for $\tau_{norm} > 2$ due to the low number of data points. In total, only 8121 non-overlapping 600 s intervals are available for 270°. For all wind turbine pairs, 11514 intervals are available as multiple winter turbine pairs are available in the same interval. The 8121 intervals resemble a measurement time of 56 days.

[Figure]

(a) Average power output fluctuation correlation per wind farm row.

[Figure]

(b) Average power output fluctuation correlation per cluster.

**Figure 1.** Comparison of the average power output fluctuation correlation for wind direction interval 270° of non-overlapping and overlapping intervals. For both plots, the average correlation curves for non-overlapping intervals are marked with 'NO' and plotted with a dashed line. The average correlation curves for overlapping intervals are plotted in both cases as solid line.

**RC12** *Figure 4: Why are the results not symmetric? I.e. for example 260 shows pronounced peak, but 280 degree does not. This aspect should be discussed. See also my above question on the selection of the 90 and 270 degrees wind directions.*

    **AC** Indeed, this effect could be caused by the asymmetric layout of the wind farm as the wind turbines are not fully horizontally aligned. Also, Fig. 4 includes all wind turbines within the wind farm. As the lower part of the wind farm shows a triangular shape while the upper part is nearly vertical, this difference in the layout could also influence the correlation curves. A more detailed study of this phenomena is not planned in this analysis. We added the following text regarding Fig. 4:

p. 9, ll. 231f: In contrast to 90°, the correlations for 270° are more defined and show slightly larger peak values. This may be due to the asymmetric wind farm layout (cf. Fig. 1). The deviation between the average correlation

curves for wind directions around 260° and 280° could be as well caused by the not entirely horizontally aligned wind turbines and by the triangular shape at the lower part of the wind farm, however, this phenomena is not further investigated in this analysis.

**RC13** *Line 203: For wind directions approaching 0° and 180° the wind turbines in a pair are oriented more perpendicular to the wind. Fluctuations reach the downstream turbine earlier. ==> If the turbines are perpendicular for these wind directions, what is then the "downstream" turbine?*

**AC** Considering an exemplary wind turbine pair where one wind turbine is labelled as wind turbine 1 and the other as wind turbine 2: we chose to set wind turbine 1 in a pair as upstream wind turbine for the 10° wind direction intervals from 0° to 170° and wind turbine 2 as upstream for wind direction intervals from 180° to 350°. As the peak of the correlations curves for wind directions around 0° and 180° is expected to be located around $\tau_{norm} = 0$, definition of upstream and downstream wind turbine does not affect the results here. This means for 90°, wind turbine 1 is be the upstream wind turbine (A) and for 270°, wind turbine 1 is the downstream wind turbine (B) and vice versa for wind turbine 2.

We added the following text:

p. 8, ll. 204f: For 10°-wind direction steps from 0° to 170° we treat the pairs according to Tab. A with reversed order and for the 10°-wind steps from 180° to 350° we treat the pairs with the given order. This means even for wind directions where both wind turbines of a pair are parallel to the wind direction, the 'upstream' wind turbine A is chosen according to the table.

And we revised:

p. 9, ll. 224f: For wind directions approaching 0° and 180° the wind turbines in a pair are oriented more perpendicular to the wind direction and fluctuations reach both wind turbines A and B at nearly the same time.

**RC14** *Line 224: Normalized by what?*

**AC** Normalised by the average power output of the upstream wind turbine A. Added in the text as follows:

p. 10, ll. 241f: Further, the power difference (normalised by the average power output of the upstream wind turbine A) and the average standard deviation of the power output fluctuations of both wind turbines in a pair are determined to analyse the flow conditions.

**RC15** *Table 2 and other tables: If possible, I believe it would also be useful to mention the average power outputs of the wind turbines A and B*

**AC** We agree this would be a valuable information. Unfortunately, absolute values power values of wind turbines are problematic in terms of confidentiality we chose to only stick to the statistics we also used for the clustering of the correlation states.

**RC16** *Line 250: "Even though such wind turbines are filtered out for the analysis, they still influence the surrounding wind turbines in an unpredictable way." ==> Please clarify the meaning of this statement. If the data is filtered such that all turbines are operations what effects are then not filtered for?*

**AC** This means that intervals where wind turbines are derated or turned off are not considered for the calculation of the correlation of power output fluctuations. However, a wind turbine that is turned off or derated influences the flow

within the wind farm and thus, the surrounding wind turbine pairs and their correlation. As example, a wind turbine could be turned off for certain time interval. This means this wind turbine is not considered for the calculation of correlations for that interval. However, a downstream wind turbine could be operating normally and be considered during that time interval. The inflow of the downstream wind turbine is affected by a higher wind speed as the flow can recover due to the turned-off upstream wind turbine. This would result in a different correlation state for the regarded wind turbine and its downstream wind turbine in comparison to other pairs in the same rows due to the affected inflow. We revised the text as follows:

> p. 13, ll. 277f: For example, if a wind turbine is turned off for a certain time interval, it is not considered in the analysis, but it still influences the flow conditions within the wind farm and the statistics or correlations of surrounding wind turbine pairs. Thus, a considered wind turbine pair downstream of the non-operating wind turbine could show a different correlation than if the upstream wind turbine would be turned on.

**RC17** *Figure 7 (line 260-263): If figure 7 is just showing two lines already shown in figure 4why do we need this additional figure?*

    **AC** Thank you for the notice, we removed Fig. 7.

**RC18** *Line 264-265: "into a reasonable set of groups and a greater number of cluster did not lead to further clusters of importance for the present analysis (see appendix B)." ==>Please clarify this statement. Also after reading the appendix this was not quite clear to me. What does "reasonable" mean? And what is a cluster of importance?*

    **AC** When grouping the data into 6 instead of 5 clusters, only clusters with similar correlation curves are found which show a slight deviation in the standard deviation of the wind turbines in a pair. For the added cluster the standard deviation of the power output fluctuations is greater for the upstream wind turbine A instead of the downstream wind turbine B. We revised the text and the appendix as follows:

> p. 13, ll. 297f: $k$ is set to five clusters. This number was empirically chosen as the data was grouped into a reasonable set of groups with clearly distinguishable correlation curves (correlation states). A greater number of clusters lead to further clusters with similar correlation curves. The only difference found was in the standard deviation of the power output fluctuations of the wind turbine pairs. Here, the cluster indicate a higher standard deviation for the upstream wind turbine A instead of the downstream wind turbine B. This slightly abnormal behaviour is shown in more detail in appendix B.

> p. 19, ll. 389f: As mentioned in Sect. 4, the number of clusters chosen for the present analysis was $k = 5$. This decision was made based on the results for $k = 6$ presented in Figure B1 and Fig. B2. For wind direction 90°, six clearly separable correlation curves are found. Comparing Fig. B1 to Fig. 7, it shows that Cluster 2 of Fig. 7 seems to be separated into to clusters (Cluster 2 and 3 of Fig. B1).
> For wind direction 270° only 5 clearly separable correlation curves are found whereas one is overlapped by a very similar one. Comparing Fig. B2 to Fig. 8, it shows that Cluster 3 of Fig. 8 seems to be separated into two similar clusters (Cluster 3 and 4 of Fig. B2). The new clusters also do not reveal any further characteristics.
> Looking at the statistics of the correlation curves listed in Tab. B1 it further can be found that for wind direction 90° Cluster 2 shows a higher standard deviation for wind turbine A instead of B while Cluster 3 shows a higher standard deviation for wind turbine B instead of A similar to all other Clusters. For wind direction 270° Cluster 4 shows a higher standard deviation for wind turbine A instead of B while Cluster 3 shows a higher standard deviation for wind turbine B instead of A similar to all other Clusters.
> The correlation curves and statistics imply that a further separation of the statistics with $k > 5$ does not reveal any correlation states which are more significant than the ones found for $k = 5$. However, clustering with $k > 5$

might result in a further distinction of flow states for wind turbine pairs based on the standard deviations of wind turbines A and B but are not further investigated as this effect is not included in the scope of the work presented here.

**RC19** *Line 308: Please be specific so the conclusion section can be read independently.*

    **AC** Thank you for noting this, the conclusion was fully revised. Please refer to the provided LaTeXDiff file.

**RC20** *Line 320: It is unclear to me what can be learned from these different clustering approaches. Why are these specific ones suggested?*

    **AC** The benefits of these algorithms have been clarified as follows:

p. 17, ll. 376f: Also, it is worth considering alternative clustering methods like $k$-medoids (Kaufman and Rousseeuw, 2008) which is less sensitive to outliers compared to $k$-means or Density-Based Spatial Clustering of Applications with Noise (DBSCAN) (Ester et al., 1996) which is also less sensitive to outliers and has no fixed cluster shapes and no predefined number of clusters like $k$-means.

**Authors response to comments from referee #2 - Mark Kelly**

General comments

**RC1** *The analysis of power output correlations across a wind farm can certainly be relevant within wind energy. This is certainly a challenging task given the data analyzed and its limitations; the latter has been only partly addressed. The abstract does not appear to include motivation for the study, and the introduction also lacks clear motivation and/or justification; why/how is this work relevant?*

    **AC** Thank you for pointing this out. We present an approach to analyse space-time correlations of the power output fluctuations of wind turbine pairs operating in a free-field wind farm. The analysis considers the effect of varying flow conditions within the wind farm and uses exclusively high-resolution SCADA data of the wind turbines. To account for these varying flow conditions, we propose a set of parameters to describe the correlations based on an approach to sort the data into groups. The space-time correlation of wind turbine pairs is for example of importance for wind turbine control and short-term load or power forecasting where the measurements from the upstream wind turbine are used to predict the wind speeds or load affecting the downstream wind turbine. We revised the abstract and conclusion carefully as well as the discussion of the results. Please see the revised manuscript with the highlighted changes.

**RC2** *Unfortunately there are a number of issues with the submitted draft, such that it requires at least major revisions. A number of plots showing pair-wise power output correlation versus normalized lag are shown; but their statistical significance is not evident, nor argued thoroughly, nor put within any context of the scales of atmospheric inflow fluctuations. (One can see by the repeated trends, of course, that the plotted $R(\tau_{norm})$ are not simply noise, despite being at most $\sim 0.1$–0.4 depending on the data selection.)*

    **AC** We agree that further that the statistical significance of the results needs further clarification. The results presented here show correlation coefficients of about 0.16 or 0.21 at first appear rather low. However, comparing these results to the findings of Lukassen et al. (2018) in the LES and of Bossuyt et al. (2017b) in the wind tunnel, it shows that our results are reasonable. The wind turbine distances, and wind speeds considered in the LES are similar to our data. The peak of the space-time correlation of the wind speed at two streamwise aligned wind turbine positions was found at about 0.5. In the wind tunnel experiments the peak of the space-time correlation of the reconstructed power output of a wind turbine in the first and second row was found at about 0.55 for aligned wind turbines and at 0.2 if the wind turbines are staggered. Here, the spacing of the wind turbines was also similar to ours and the thrust coefficients of the discs resemble operation below rated wind speed. Thus, considering the varying flow conditions in our data set, the peaks of about 0.16 or 0.21 are rather good in comparison to the results in the steady flow conditions.
We clarified the relevance of the presented correlations as follows:

> p. 8, ll. 218f: The maximum correlation around 0.2 may seem rather low but is reasonable considering the high variability in the flow and wind turbine dynamics in free field measurements. As comparison, in the LES study of Lukassen et al. (2018), a maximum correlation coefficient of about 0.5 was found for space-time correlations of wind speeds measured at comparable distances with comparable wind speed. In the wind tunnel experiments by Bossuyt et al. (2017b), a maximum correlation of about 0.55 was found for the space-time correlation of the reconstructed power output of discs placed at comparable distances with comparable wind speeds.

> p. 17, ll. 357f: For 90° the peak of the correlation increased via clustering from 0.16 to 0.32 and for 270° the peak of the correlation increased from 0.21 to 0.41. A value of 0.41 is close to the correlations found in the LES study by Lukassen et al. (2018) and experiments by Bossuyt et al. (2017b) which were between 0.5 and 0.55 for similar wind turbine spacing and similar wind speeds.

**RC3** *The methods used are not explained in sufficient detail, with references to such also lacking. As currently reported, the study would not be reproducible by a reader. The results need to be more clearly presented, and also interpreted, by the authors—in addition to the inclusion of relevant details, motivation, and significance of the study.*

> **AC** We agree and clarified the methods, especially in Sect. 2.2. Further we clarified the description and discussion of the results. Please see the revised manuscript with the highlighted changes.

**RC4** *There are errors in language throughout; I have included some correction examples in the last section below, but suggest that the next version be proofread by somebody with near-native level fluency.*

> **AC** Thank you for pointing this out. We proofread the paper and corrected the linguistic errors. Please see the revised manuscript with the highlighted changes.

General comments

**RC5** *l.8: what is meant by 'correlation states'? Lines 7–9, starting with 'Most importantly', should be reworded. To be more direct: you are using a clustering algorithm (k-means clustering) to group similarly correlated turbine pairs, in order to examine the spatial variation of correlations between turbines and determine the key parameters affecting such correlations.*

> **AC** Correlation states are correlations caused by varying flow conditions which are found for certain wind turbine pairs within the wind farm based on the introduced parameters: standard deviations of the power output fluctuations and the power difference of wind turbine in a pair. The abstract was revised as follows:

> p. 1, ll. 1f: Space-time correlations of wind turbine pairs provide information on the flow conditions within a wind farm and on the interactions of the wind turbines. Such information plays an important role for the control of wind turbines and short-term load or power forecasting. However, the challenge to analyse space-time correlations of power output fluctuations of wind turbine pairs in a free field wind farm are the highly varying flow conditions. Here, we present an approach to investigate space-time correlations of power output fluctuations of wind turbine pairs in free field based on high-resolution SCADA data, which overcomes the challenge of highly variable flow conditions within the wind farm. Using eight months measurements from an offshore wind farm with 80 wind turbines, the influences of different parameters on the correlation of power output fluctuations are analysed. Wind direction investigations show that correlations of power output fluctuations of wind turbine pairs are highest for streamwise aligned pairs and decrease towards spanwise pairs. Further, it is found that the correlation of power output fluctuations of streamwise aligned wind turbine pairs depends on the location of the wind turbines within the wind farm as well as the inflow conditions (free-stream or wake). The main outcome is that the correlation of streamwise aligned wind turbine pairs can be characterised by the standard deviations of the power output fluctuations and the power difference of the wind turbines in a pair. Evaluating these parameters with the data-driven clustering algorithm *k*-means, wind turbine pairs with similar power output fluctuation correlations are grouped depending on these parameters and independent from their location. These groups are here referred to as correlation states. With this approach we account for the highly variable flow conditions inside a wind farm

which influence the correlations in an unpredictable way. As a final result we shows that these parameters lead to clearly distinguishable correlation states.

**RC6** *l.9–10: The phrase 'next to' isn't appropriate here; it seems you're wanting to say 'in addition to' or similar. Also, do you mean the location of a turbine pair is most important, or the relative locations/distance?*

    **AC** The location of a wind turbine pair within the wind farm is most important. Please find the revised abstract in **RC5**.

**RC7** *l.38: in terms of correlations, the wind farm was not "infinitely large", was it? I.e., were there not periodic boundary conditions used in the LES?*

    **AC** Yes, periodic boundary conditions were used. The text was revised for clarification as follows:

p. 2, ll. 44f: In an LES study by Lukassen et al. (2018) velocity space-time correlations within a wind farm with periodic boundary conditions (modelling a periodic array of wind turbines) were analysed and modelled analytically.

**RC8** *l.40: by 'variance of the wind velocity', do you not mean strength of turbulence in the prescribed inflow?*

    **AC** If you refer to the turbulence intensity: both parameters, the variance of the wind velocity and the mean wind velocity in the inflow enter the model as explicit parameter and not as the combined parameter turbulence intensity. To clarify this, the text was revised as follows:

p. 2, ll. 45f: The velocity fluctuations, which are directly related to power output fluctuations showed pronounced space-time correlations. Furthermore, the variance of the wind velocity and the mean velocity turned out to be important parameters in the modelling set up.

**RC9** *l.55,l.64 and elsewhere: you haven't (yet) defined "correlation state"*

    **AC** Thank your indicating this. We define correlation states the average correlations found by the clustering of the statistical parameters: the standard deviations of the power output fluctuations and the power difference of wind turbine in a pair. The definition of correlation states was added as follows (also see **RC5**):

p. 2, ll. 60f: In this paper we investigate the influencing factors on the correlation of power output fluctuations of wind turbine pairs and introduce parameters to distinguish different correlation curves, here called correlation states. These parameters are then evaluated with a data-driven clustering algorithm with the aim to group the data according to the underlying correlation states.

**RC10** *l.71: why "non-axisymmetric"? Do you mean asymmetric?*

    **AC** Yes, thank you. We corrected the sentence as follows:

p. 3, ll. 78f: They are installed in a grid like, slightly asymmetric pattern with a triangular shape towards south (see Fig. 1).

**RC11** *l.74-76 and afterward: How did you calculate the wind direction and U? What transfer function(s) were used, and how was this blended with speed implied by measured power?*

    **AC** Unfortunately, $U$ is provided as a variable within the data set. It is not measured but directly related to the measured power and reconstructed from the wind turbine (controller) settings. Details on the reconstruction are not available. Depending on $U$, the correlation curves stretch or shrink (see p.6, eq. 3). Thus, if the recalculated wind speed differs from the actual wind speed, the normalised correlation curve might be slightly shifted to a larger $\tau_{norm}$. However, $U$ affects the normalisation of the correlation curves in a consistent manner for all wind turbines. For this reason, we consider $U$ as reasonable variable in the context of this analysis. We revised the description of $U$ and how it affects the analysis:

> p. 3, ll. 81f: The processed signals include the generated power $P$, the azimuth angle of the wind turbines (i.e. the nacelle direction) $\theta$, the nacelle based wind direction $\varphi$ (measured relative to $\theta$), the pitch angle $\beta$ of each blade, and a reconstructed wind speed $U$.
>
> The reconstructed wind speed $U$ is not directly measured but provided as a variable which results from the measured power and control variables of the wind turbine. Due to that, $U$ is considered as an approximated and idealised value which does not include wind speed independent power reduction, e.g. by misalignment of the wind turbine due to measurement errors of the wind direction. In the context of this work, it can still be used for assessing the effect of the wind speed on the correlations of power output fluctuations of wind turbine pairs which is further discussed in Sect. 2.2.

> p. 7, ll. 179f: As mentioned before, $U_B$ is reconstructed and might differ from the actual wind speed affecting the wind turbines. However, in the context of this normalisation the effect on the resulting correlations curves is marginal as the correlation curves may only be slightly shifted due to the deviation to the real wind speed.

**RC12** *l.77: do you mean that the yaw error was not used in the nacelle wind speed transfer function/correction?*

    **AC** The yaw measured by the wind vane could be taken into account for the reconstruction of the wind speed. However, a wind turbine could have a further unknown yaw misalignment which does not show in the data as it is caused by false measurements, inaccurate or false sensor calibrations or false sensor installation. In this case the wind turbine would have a greater yaw than the measured one and it would measure less wind speed due to that. As mentioned in **RC11**, unfortunately, we have no further information on the reconstruction.

**RC13** *l.87–89: did you consider the across-farm variation in wind direction, with suitable averages of upwind turbines?*

    **AC** No, we did not consider the across-farm variation in wind direction. The mentioned inaccuracies in the $\theta_i$ and $\varphi_i$ measurements (l. 92-98) prevent a precise evaluation of different wind directions within the wind farm, i.e. the wind directions at certain wind turbines. Thus, we average the wind direction over all available wind turbines within the wind farm considering the effect of certain wind turbines facing another direction. Please see the description of the azimuth angle $\theta$ and the description of the average wind direction $\varphi_{av}$:

> p. 3, ll. 89f: The azimuth angle $\theta$ of the wind turbine refers to the direction it is facing in its preset reference system. This system does not necessarily exactly match to the global geographical one due to the measurement inaccuracies of the azimuth angle and a potentially inaccurate north orientation of the reference system of each wind turbine (cf. Bromm et al., 2018).
>
> The nacelle based wind direction $\varphi$ is estimated based on the measurements of two 2D sonic anemometers installed behind the rotor of each wind turbine. These measurements have to be treated with care as the measured flow behind the rotor is disturbed by the rotation of the rotor and the nacelle itself. Thus, it is only an estimation of the wind direction and yaw of the wind turbine. However, as shown by Dai et al. (2017), wind direction fluctuations at reasonable yaw angles ($< 45°$) have only little effect on the power output fluctuations of wind turbines and thus inaccuracies in $\varphi$ have no major influence on the performed analysis. The combined measurements of $\theta_i$ and $\varphi_i$ define the wind direction $\Phi_i$ at the $i$-th wind turbine.

p. 4, ll. 98f: For assessing an average wind direction for the wind farm, we average over $\Phi_i$ of all wind turbines to reduce the influence of false measurements of single wind turbines. Due to the size of the considered wind farm, the wind direction is not expected to be consistent throughout the whole wind farm. Single wind turbines could be facing different wind directions compared to the average wind direction of the wind farm (cf. Schneemann et al., 2020; Sanchez Gomez and Lundquist, 2020). The wind direction of the wind farm averaged over all available wind turbines is defined as $\Phi_{av}$.

**RC14** *l.111 and Table 1: how did you determine the pitch threshold?*

**AC** This value was empirically chosen based on the analysed data set. In this case, this was the threshold for non-pitched blades. For clarification we added the following text:

p. 5, ll. 122f: Hence, to fully exclude pitching wind turbines, the data is filtered for any pitching activity. Please note that for this specific data set this implies that $\beta < -1.3°$.

**RC15** *l.112–116 and Table 1: what is the yaw misalignment threshold? What about $\varphi$ change per 600s?*

**AC** There is no threshold for yaw misalignment within the 600 s intervals that means no yawing at all. The maximum possible misalignment is thus defined by the controller settings of the wind turbines which define at what yaw misalignment the wind turbine starts yawing. As the wind vane measurements are highly dynamic within the considered 600 s intervals a limitation of $\varphi$ would lead to a significant loss in the amount of considerable correlation intervals as nearly no intervals with consecutively measured values would be available after filtering.

**RC16** *l.120–121: why not report the variation of angles, instead of only the mean with 10-degree tolerance?*

**AC** Within the scope of this work we introduce an approach to analyse correlations of power output fluctuations of wind turbine pairs which copes with the varying flow conditions within the wind farm. An investigation of the influence of the misalignment of individual wind turbines from the average wind direction of the wind farm is not part of this analysis and thus not further taken into account.
Also, Dai et al. (2017) presented that wind direction fluctuations of wind turbines have only a small influence on the power output fluctuations of (yawed and aligned) wind turbines. Please see the text including description of the azimuth angle $\theta$ in **RC11**.

**RC17** *l.155–159: mention the use of Taylor's hypothesis here—and the assumption you're thus using over the entire range of $\tau_{norm}$. There are several references you should check (and cite) regarding this.*

**AC** Thank you for your notice. Indeed, our approach is similar to Taylor's hypothesis. However, we do not further investigate it in this context. We revised the text as follows:

p. 6, ll. 166f: Similar to Taylor's hypothesis (Taylor, 1938) we assume that depending on the wind speed, wind structures responsible for power output fluctuations measured at an upstream wind turbine A, take some time to travel the distance to the downstream wind turbine B.

**RC18** *l.163–166: Your statement about $\tau_{norm} > 1$ has an implication: if you used the average speed between turbines A and B (instead of $U_B$), then wouldn't $\tau_{norm} = 1$, unless the propagation speed is somehow otherwise affected? It's difficult to defend using just $U_B$*

    **AC** Yes, this is correct. We stick to $U_B$ as reference, since the average advection speed between the wind turbines is not available. The only available measurements are the wind speeds at the upstream wind turbine A and the downstream wind turbine B. As the upstream wind turbine mostly affected by higher wind speeds than the downstream wind turbine B, we choose $U_B$ as reference. Of course, $U_B$ is also an overestimation of the actual average speed between wind turbine A and B. However, $U_B$ shrinks or stretches the correlation curves and an overestimation slightly shifts the normalised correlation curves towards larger $\tau_{norm}$. This effect is consistent on all wind turbines and thus we consider the usage of $U_B$ as reasonable. We revised Sect. 2.2 to clarify this dependency:

> p. 3, ll. 81f: The processed signals include the generated power $P$, the azimuth angle of the wind turbines (i.e. the nacelle direction) $\theta$, the nacelle based wind direction $\varphi$ (measured relative to $\theta$), the pitch angle $\beta$ of each blade, and a reconstructed wind speed $U$.
>
> The reconstructed wind speed $U$ is not directly measured but provided as a variable which results from the measured power and control variables of the wind turbine. Due to that, $U$ is considered as an approximated and idealised value which does not include wind speed independent power reduction, e.g. by misalignment of the wind turbine due to measurement errors of the wind direction. In the context of this work, it can still be used for assessing the effect of the wind speed on the correlations of power output fluctuations of wind turbine pairs which is further discussed in Sect. 2.2.

> p. 7, ll. 175f: Due to this definition of $\tau_{norm,intv}$ and $\tau_{norm}$ (see Eq. **??**), the peak of the correlation curves is expected to be found at $\tau_{norm} = 1$ if the advection speed of the wind speed fluctuations matches the wind speed affecting B. Thus, in partial load situations where wind turbine B is in the wake of wind turbine A, the peak is expected to be at $\tau_{norm} > 1$. Here, the reduced wind speed in the wake recovers slowly, so that the wind speed affecting wind turbine B, i.e., $U_B$ is already partly recovered and hence larger than the advection speed of the fluctuations. As mentioned before, $U_B$ is reconstructed and might differ from the actual wind speed affecting the wind turbines. However, in the context of this normalisation the effect on the resulting correlations curves is marginal as the correlation curves may only be slightly shifted due to the deviation to the real wind speed.
> In a next step, the correlation curves with the normalised lag $\tau_{norm,intv}$ are discretised using a histogram with a reference time lag of
>
> $$\tau_{norm} = \tau \cdot \frac{U_{max}}{x_{AB,mean}} \qquad (4)$$
>
> where $\tau$ is the time lag (0 s to 300 s), $U_{max}$ is an artificially introduced velocity which has to be at least equal to the maximum possible wind speed to fit all normalised curves (here $U_{max} = 13 \text{ ms}^{-1}$). $x_{AB,mean}$ is the average distance between wind turbine A and wind turbine B of the considered wind turbine pairs. Note that $\tau_{norm,intv}$ is only used for stretching and shrinking of the correlation curves and that $\tau_{norm}$ is used only for binning of the stretched or shrunk correlations.

**RC19** *l.181: why 20deg interval? How does this compare to the variation of $\varphi$ across all turbines for a 10-minute periods? The latter is likely important to describe the inflow state and variability of the wind field (not to mention yaw error).*

    **AC** We investigate the average correlation of all wind turbine pairs in the wind farm for different main wind directions. These main wind directions are the wind directions averaged over the whole wind farm. The 20° interval is the result of the 10-degree tolerance in the wind direction measurements of the wind turbines. In Fig. 3 we present the average correlation of all wind turbine pairs in the wind farm for the corresponding average wind farm direction. As

mentioned before, in the scope of this work we do not investigate the influence of wind direction fluctuations or the influence of wind turbine misalignments on the correlation of wind turbine pairs.

**RC20** *p.8/l.201–203: Indeed $\rho = 0.2$ is not generally considered to be statistically significant. Have you tried windows of different lengths (other than 300s)? Have you considered the integral timescale of the incoming turbulence? What about the wake turbulence?*

> **AC** Yes, a correlation coefficient of 0.2 is generally not considered to be statistically significant, however considering the effect of the high dynamics in the flow of free-field measurements it is reasonable. Especially when comparing these findings to the LES simulations of Lukassen et al. (2018) where wind speed space-time correlations at one wind turbine distance showed a correlation coefficient of 0.5. In the wind tunnel experiments of Bossuyt et al. (2017b) a correlation of about 0.55 was found for the space-time correlation of wind turbines with one wind turbine distance. This comparison is also clarified in the paper:

> p. 8, ll. 218f: The maximum correlation around 0.2 may seem rather low but is reasonable considering the high variability in the flow and wind turbine dynamics in free field measurements. As comparison, in the LES study of Lukassen et al. (2018), a maximum correlation coefficient of about 0.5 was found for space-time correlations of wind speeds measured at comparable distances with comparable wind speed. In the wind tunnel experiments by Bossuyt et al. (2017b), a maximum correlation of about 0.55 was found for the space-time correlation of the reconstructed power output of discs placed at comparable distances with comparable wind speeds.

> The integral timescale was not investigated, but it can be stated that the temporal autocorrelation of a wind turbine decorrelates in the considered time intervals of 300 s.
> Further, the wake turbulence has not been taken into account in detail. However, it is resembled in the standard deviations of the power output fluctuations which are introduced as parameter. The cluster results clearly show the influence of the standard deviation on the correlations of wind turbine pairs. The highest correlation is found for the cluster with the highest standard deviations.

**RC21** *Table 3: is it reasonable to include 3 significant digits in the correlations listed? (or 5 in the power?) Such second-order statistics do not converge so easily within 10 minutes...*

> **AC** Thank you for pointing this out. We adjusted Tab. 2 to 4 and B1.

**RC22** *l.253–255: While this reviewer has some familiarity with k-means clustering, how is a reader supposed to be able to understand (let alone repeat) the analysis reported here? Please include appropriate references and details.*

> **AC** We agree that further information is necessary, thus a short description of the $k$-means algorithm was added:

> p. 13, ll. 284f: $k$-means is an algorithm which iteratively sorts data into $k$ clusters. After choosing an initial centre for each cluster (centroids) within the data, all data points are assigned to their nearest centroid. Afterwards, the new centres of the clusters are calculated based on the assigned data points. These steps are repeated until a previously defined number of iterations is reached or the centres of the clusters no longer change. Finally, the data is distributed into $k$ clusters. The result of $k$-means is dependent on the starting positions of the cluster centres. Thus, the algorithm can be repeated with changing starting points for the clusters to find the best possible solution.

> The details of the performed clustering are given here:

> p. 13, ll. 293f: The clustering is performed using the $k$-means algorithm of MATLAB (MATLAB, 2019) based on Lloyd (1982), using random sample points as initial centroids to find the best solution. To avoid the generation

of local centroids the clustering is repeated ten times and the run with the clusters with the lowest sum of point-to-centroid distances within the clusters is chosen. As a distance metric for the clusters, the squared Euclidean distance chosen. The maximum number of iterations is set to 300.

**RC23** *l.293 and elsewhere: instead of just 'filtering', perhaps you should use a term like 'data selection process'; recall that in the spectral sense a filter means something else, and such filtering could be expected for the kind of analysis you do here.*

    **AC** We agree that 'filtering' could refer to something else dependent on the context. The term 'data filtering' or 'filtering of data' is commonly used and known in data science. Based on this we came to choose the term 'filtering' in this context.

**RC24** *l.296: "Deviations from the 90° and 270° wind directions result in a decreased correlation" is out of context here; this and the statement after it do not make sense as written.*

    **AC** We agree, this issue is resolved in the revised conclusion.

**RC25** *l.299–300: pairs within the first (upwind) row of turbines have little correlation not simply because of the 'free-stream' inflow, but because their spacing is greater than the transverse integral length scale of the turbulence.*

    **AC** With pairs within the first row we were referring to streamwise aligned pairs where the upstream wind turbine A is located in the very first row and the downstream wind turbine B is located in the second row of the wind farm. To clarify this, we added and revised the sentence:

> p. 16, ll. 344f: Wind turbine pairs, where upstream wind turbine A is located in the first row and downstream wind turbine B is in the second row of the wind farm, show no correlation with large normalised power differences and small standard deviations of power output fluctuations. This is most likely caused by the free-stream inflow of the upstream wind turbines A of the pairs.

**RC26 RC23:** *l.308–309: simply using 'previously defined' or 'chosen statistics' is somewhat obfuscatory and not really appropriate in a conclusion/summary; it does not explain to the reader which 'statistics' you are considering.*

    **AC** We absolutely agree with this and resolve this issue in the revised conclusion.

**RC27** *l.309–311: why not just use joint distributions, and conditional statistics? How is kmeans more helpful?*

    **AC** Considering the three introduced parameters, the evaluation of joint distributions is more complex as more intermediate steps are needed. In addition, the data has to be grouped manually to average the correlation curves. *K*-means on the other hand performs everything in one run and is easily adaptable to further parameters.

**RC28** *l.311–312: this statement is not understandable without more context; please help the reader, and interpret it also.*

    **AC** We agree, this issue is resolved in the revised conclusion:

> p. 17, ll. 353f: The clustering showed similar results for wind directions 90° and 270° and the clusters showed clearly distinguishable parameters which were directly related to the average correlation curve per cluster. Increased standard deviations combined with small power differences showed the most defined correlations with the highest peak. This combination was found for wind turbine pairs with a position more downstream in the wind farm but also including some wind turbine pairs from rows towards the front. For 90° the peak of the correlation increased via clustering from 0.16 to 0.32 and for 270° the peak of the correlation increased from 0.21 to 0.41. A value of 0.41 is close to the correlations found in the LES study by Lukassen et al. (2018) and experiments

> by Bossuyt et al. (2017b) which were between 0.5 and 0.55 for similar wind turbine spacing and similar wind speeds.

**RC29** *l.320–323: why were no details about this given in section 4?*

   **AC** This is discussed in the revised appendix and the following text was added:

> p. 13, ll. 297f: $k$ is set to five clusters. This number was empirically chosen as the data was grouped into a reasonable set of groups with clearly distinguishable correlation curves (correlation states). A greater number of clusters lead to further clusters with similar correlation curves. The only difference found was in the standard deviation of the power output fluctuations of the wind turbine pairs. Here, the cluster indicate a higher standard deviation for the upstream wind turbine A instead of the downstream wind turbine B. This slightly abnormal behaviour is shown in more detail in appendix B.

Technical corrections

**RC30** *l.6: 'correlation is' should be 'power correlations are'; 'towards' is not appropriate*

   **AC** Revised.

**RC31** *l.15: second 'and' should be 'while' or similar.*

   **AC** Corrected.

**RC32** *l.23: 'the respective' should be 'a respective'*

   **AC** Corrected.

**RC33** *l.27–29: need to re-word run-on sentence, including e.g. appropriate commas*

   **AC** Corrected.

**RC34** *l.33: remove 'a' before 'high'*

   **AC** Corrected.

**RC35** *l.46: 'of' should be 'between'*

   **AC** Corrected.

**RC36** *l.81: reference missing year*

   **AC** Corrected.

**RC37** *l.400: incorrect journal and doi for Valldecabres reference*

   **AC** Corrected.

**RC38** *l.403–404: the WindEurope reports appear to be missing some identifying information (e.g. doi, report number, etc.).*

   **AC** Revised.

**References**

Bossuyt, J., Howland, M. F., Meneveau, C., and Meyers, J.: Measurement of unsteady loading and power output variability in a micro wind farm model in a wind tunnel., Experiments in Fluids, 58, 1–17, https://doi.org/10.1007/s00348-016-2278-6, 2017a.

Bossuyt, J., Meneveau, C., and Meyers, J.: Wind farm power fluctuations and spatial sampling of turbulent boundary layers, Journal of Fluid Mechanics, 823, 329–344, https://doi.org/10.1017/jfm.2017.328, 2017b.

Bromm, M., Rott, A., Beck, H., Vollmer, L., Steinfeld, G., and Kühn, M.: Field investigation on the influence of yaw misalignment on the propagation of wind turbine wakes, Wind Energy, 21, 1011–1028, https://doi.org/10.1002/we.2210, 2018.

Dai, J., Cao, J., Liu, D., Wen, L., and Long, X.: Power fluctuation evaluation of large-scale wind turbines based on SCADA data, IET Renewable Power Generation, 11, 395–402, https://doi.org/10.1049/iet-rpg.2016.0124, 2017.

Ester, M., Kriegel, H.-P., Sander, J., and Xu, X.: A Density-Based Algorithm for Discovering Clusters in Large Spatial Databases with Noise, in: Proceedings of the Second International Conference on Knowledge Discovery and Data Mining, KDD'96, p. 226–231, AAAI Press, 1996.

Kaufman, L. and Rousseeuw, P.: Partitioning Around Medoids (Program PAM), chap. 2, pp. 68–125, John Wiley & Sons, Ltd, https://doi.org/10.1002/9780470316801.ch2, 2008.

Lloyd, S. P.: Least squares quantization in PCM, IEEE Transactions on Information Theory, 28, 129–137, https://doi.org/10.1002/9780470316801.ch2, 1982.

Lukassen, L. J., Stevens, R. J. A. M., Meneveau, C., and Wilczek, M.: Modeling space-time correlations of velocity fluctuations in wind farms, Wind Energy, 21, 474–487, https://doi.org/10.1002/we.2172, 2018.

MATLAB: version 9.7.0.1190202 (R2019b), The MathWorks Inc., Natick, Massachusetts, 2019.

Ramírez, L., Fraile, D., and Brindley, G.: Offshore Wind in Europe - Key trends and statistics 2019, https://windeurope.org/wp-content/uploads/files/about-wind/statistics/WindEurope-Annual-Offshore-Statistics-2019.pdf, last access 21.01.2021, 2020.

Sanchez Gomez, M. and Lundquist, J. K.: The effect of wind direction shear on turbine performance in a wind farm in central Iowa, Wind Energy Science, 5, 125–139, https://doi.org/10.5194/wes-5-125-2020, 2020.

Schneemann, J., Rott, A., Dörenkämper, M., Steinfeld, G., and Kühn, M.: Cluster wakes impact on a far-distant offshore wind farm's power, Wind Energy Science, 5, 29–49, https://doi.org/10.5194/wes-5-29-2020, 2020.

Taylor, G. I.: The Spectrum of Turbulence, Proceedings of the Royal Society A: Mathematical, Physical and Engineering Sciences, 164, 476–490, https://doi.org/10.1098/rspa.1938.0032, https://royalsocietypublishing.org/doi/pdf/10.1098/rspa.1938.0032https://royalsocietypublishing.org/doi/10.1098/rspa.1938.0032, 1938.

---

## Referee Report (RR1)

**Mark Kelly**

The draft has been improved with revision; however, some issues still remain. These are pointed out below, with author comments (AC) addressed first, and later point-wise comments with line numbers referring to the file that showed revision/changes ("...ATC1.pdf").

There are also numerous linguistic errors which need to be corrected (most of them new). As in the previous review, I again suggest asking someone with native-level English proficiency to proof-read the (updated) draft; they are too numerous to note individually here.

E.g., on l.6: "in free field" is not proper English; on l.8 the word 'of' is missing after "eight".

Before getting to specific points line-by-line, I'll respond to the author comments (ACs) that addressed my previous reviewer comments (RCs), and then I add some general comments.

**Replies to author comments/responses (ACs)**

AC11: It is an improvement to mention the "reconstructed" aspect of wind speed, but to be open/clear, why not include your AC statement "details on the reconstruction are not available" in the text? This and/or the unknown transfer function should be mentioned; e.g. the latter can affect the direction as well as the speed.

AC12: Yaw misalignment isn't only due to "false...measurements, calibrations, or sensor installation"–especially if there is a transfer function used for nacelle-mounted 2d-anemometers. Your addition on l.106-8 helps to allay this issue.

AC15: Your response about yaw misalignment threshold and rate per 10-minutes is reasonable, but you have not included this in the revised text.

AC17: How is your approach "similar to" Taylor's, but not actually simply assuming it? Again, it appears you've assumed it over the entire range of $\tau$ (and $\tau_{norm}$); this can become problematic for small enough $U$ (large lags).

AC18 and l.189-205: you have used $\tau_{norm}$ before it is defined in (4); this can be quite confusing for the reader, particularly if they have not read this before. Also, why is the second 'normalization' (4) done? If $U_{max} = 13$m/s always, and $x_{AB,mean}$ is also a simple constant for all cases, then why normalize again? If you have done this to force the peaks closer to 1, then this

should be stated. Also, how was $U_{max}$ chosen–doesn't this just arbitrarily squeeze/stretch the correlation curves (as you wrote for $\tau_{norm,intv}$)? There appears to be no physical justification for (3) and (4) together, unless perhaps you could explain what is meant by "at least equal to the maximum possible wind speed to fit all normalised curves".

AC20: The response statement "the temporal autocorrelation of a wind turbine decorrelates in the considered time intervals of 300 s" does not make sense. Do you perhaps mean that the the correlation decreases to effectively 0 as lag ($\tau$) approaches 300s?

AC23: If you are to insist on using the term "filtering" in place of 'data selection' or similar—knowing that WES is not a data science journal, but a wind energy journal where you are also mentioning turbulence—then you should at least include the word 'data' before it. Further, I strongly recommend section 2.1 to be renamed "Data selection and filtering" or similar—again, spectral filtering is commonly used when dealing with this kind of data in wind energy (particularly turbulence), especially when mentioning different intervals (e.g. 600s).

**General comments**

The labelling of peak correlations (between power fluctuations for turbine pairs) as "correlation states" is contentious, since 'states' implies different physical scenarios or flow/operational-regimes—particularly if you have not described anything like the latter. If the 'states' are basically different groups of peak correlations (magnitudes), then why not call them that? This is safer, because for different flow regimes having larger/smaller turbulence length scales (and/or other farms having different spacing, surfaces, or even hub height), then the magnitudes or groups could be quite different. Also, in the conclusion it should be mentioned how/why seemingly insignificant peak correlations (e.g. 0.2 or less) are meaningful (compared to commonly understood statistical significance being $\rho_{AB} > 0.5$); i.e. the relative values are significant given the 'noisy' turbulent flow, in addition to values consistent with others' LES results.

**Some detailed comments**

l.5, 7: the challenge is *spatially* variable flow, not just "highly" variable flow.

l.10-11: "decrease towards spanwise pairs" doesn't quite make sense. If the correlations decrease with angle between mean wind direction and pair separation vector, why not write that?

l.13-17: "the correlation of streamwise aligned wind turbine pairs" should be 'power correlations between streamwise-aligned wind turbine pairs'.

l.18: sorting is accomplished by a "k-means clustering algorithm", not "clustering algorithm k-means" (e.g. Likas, Vlassis, & Verbeek 2003).

l.19-20: the sentence "These groups are here referred to as correlation states." is not needed in an abstract.

l.20 repeats l.7-8.

l.18-22: "these parameters" is repeated three times; the final point is also somewhat of a repetition of l.16-17...The abstract can be cleaned up (there are more English errors in it as well).

l.71: Use of "correlation curves" here to mean "states" is ambiguous and confusing; however, in l.194 and after it is used reasonably, to refer to the actual $R(\tau)$ curves. Here in this context of "states" you are really referring to the peak correlation, as seen later in e.g. Figs.7-8.

l.151: sentence is not finished.

l.196: "noted as" $->$ 'denoted by'

l.212 and elsewhere: "dependency" should be 'dependence'

l.242: how does $\tau_{norm} > 1$ have any meaning, if $\tau_{norm}$ is arbitrary due to its definition via the artificial Umax?

l.381: exactly which standard deviation?

---

## Author Response (AR2)

**Correlations of power output fluctuations in an offshore wind farm using high-resolution SCADA data**

Authors: Janna K. Seifert, Martin Kraft, Martin Kühn, and Laura J. Lukassen

DOI: 10.5194/wes-2020-96
* * *
**Authors response to referee comments**

Dear referees, we appreciate the time you again spend helping to improve our manuscript. Your constructive comments help us to improve our paper further. As recommended, we had our manuscript professionally proofread by a native speaker. Below we will discuss your comments in detail.
* * *
**Authors response to comments from referee #1**

**RC1** *Overall I believe this line of work is important and relevant for the community and based on this I believe this work is suitable for publication in Wine Energy Science. The authors have replied in detail to the comments of the referees and the corresponding changes have improved the manuscript. Obviously, the analysis of the field data is a challenging task as articulated by the authors. I believe that in various places, as indicated below, the presentation in the manuscript can be further clarified.*

**AC** Thank you for your positive feedback. We appreciate your effort and will incorporate your remarks as follows.

Comments on previous review

**Referee #1**

**RC2** *RC4: So to clarify; the data on U is provided to you (and is based on Power signal) and you did not calculate it yourself.*

**AC** Yes, that is correct. $U$ is not calculated by us but provided as is without detailed information.

**RC3** *RC6: The slight asymmetry is mentioned in the conclusions (line 340), but I am not sure it is mentioned clearly in the body of the text, please double check.*

**AC** We double checked the text and we mention the asymmetry first here l. 83f: "They are installed in a grid-like, slightly asymmetric pattern with a triangular shape towards the south (see Fig. 1)." And again here l.252f: "Figure 4 shows the average power output fluctuation correlation around 90° and 270° in detail as cuts through Fig. 3. The absolute peaks are at 90° and 270°. For wind directions where the wind turbines in a pair are less streamwise aligned, the peak decreases, and the correlation curve flattens. In contrast to 90°, the correlations for 270° are more defined and show slightly larger peak values. This may be due to the asymmetric wind farm layout (cf. Fig. 1).

**RC4** *The information provided as response to RC11 is very useful, but I could not find this information in the manuscript. I think the answer can be summarized shortly in the manuscript, and the authors can refer to the graph provided in the response document, which is already available online. So the graph itself does not need to be incorporated in the manuscript. But it is useful to inform the reader of the main manuscript that it is available.*

**AC** We agree that this information is helpful for the reader and decided to include it in the appendix of the paper to enable the reader to check on the information directly. In the text we added:

> p.6, ll. 154f: Depending on the data availability, the next interval of 600 consecutive seconds could go from $[t_j + 1\,\text{s}, t_j + 1\,\text{s} + 599\,\text{s}]$, and thus overlap the previous one up to 599 s. This does not result in significantly different findings compared to non-overlapping intervals as shown in App. B.

**RC5** *RC20: the provided answer does not really answer my question. The provided answer states the general benefit of the approach, but I am wondering whether the authors can clarify the specifically new insights obtained for the case considered here.*

    **AC** To clarify further clarify this, the other cluster approaches could help to find more defined correlation curves (higher and more narrow peaks) or to find further clusters. We revised the text as follows:

> p. 18, ll. 409f: In addition, it is worth considering alternative clustering methods like *k*-medoids (Kaufman and Rousseeuw, 2008), which is less sensitive to outliers or Density-Based Spatial Clustering of Applications with Noise (DBSCAN) (Ester et al., 1996) which is also less sensitive to outliers and has no fixed cluster shapes and fixed number of clusters. Using these algorithms could improve the clustering of the intervals and more defined correlation curves or could identify further clusters.

**Referee #2**

**RC6** *RC19: The authors state "The 20° interval is the result of the 10-degree tolerance in the wind direction measurements of the wind turbines". I think this information is very important and it should be incorporated in the main text.*

    **AC** We agree. Thank you for pointing this out. We revised the text as follows:

> p. 8, ll. 209f: In the following, we average correlations over a wind direction interval of 20° and all available time intervals of the considered wind turbines (either all wind turbines or a selection of wind turbines). We consider 20° intervals due to a 10° tolerance in the wind direction measurements of the wind turbines.

Specific comments

**RC7** *I also read the manuscript again. In various places the authors could be more precise on the formulation, i.e. making clear to what case, data, parameter is exactly referred. Or the English grammar should be removed. I provided a list of some of the main instances below, but please check the entire manuscript, especially the abstract, on this.*

    **AC** We agree and revised the text accordingly. Please see the track of changes as well as the answers to your specific comments below.

**RC8** *"Space-time correlations of wind turbine pairs" ==> the power output is correlated; not the turbine pairs*

    **AC** Thank you for pointing this out. Revised as follows:

> p.1, ll. 1f: Space-time correlations of power output fluctuations of wind turbine pairs provide information on the flow conditions within a wind farm and the interactions of wind turbines.

**RC9** *"Such information plays an important role for the control" ==> Can provide important insights for controls, i.e. information obtained from the analysis you presented is not yet used in wind farm control*

    **AC** Thank you for noting this. The text was revised as follows:

> p.1, ll. 2f: Such information can play an essential role in controlling wind turbines and short-term load or power forecasting.

**RC10** *which overcomes the challenge of highly variable flow conditions within the wind farm ==> make this a new sentence. New it refers to the SCADA data. However, this statement is on your analysis approach.*

    **AC** We agree, the text was revised as follows:

> p.1, ll. 4f: Here, we present an approach to investigate space-time correlations of power output fluctuations of streamwise-aligned wind turbine pairs based on high-resolution SCADA data. The proposed approach overcomes the challenge of spatially variable and temporally variable flow conditions within the wind farm. We analyse the influences of the different statistics of the power output of wind turbines on the correlations of power output fluctuations based on eight months of measurements from an offshore wind farm with 80 wind turbines.

**RC11** *Wind direction investigations show ==> More effect of the wind direction.*

    **AC** That is correct. The text was revised as follows:

> p.1, ll. 8f: First, we asses the effect of the wind direction on the correlations of power output fluctuations of wind turbine pairs. We show that the correlations are highest for the streamwise-aligned wind turbine pairs and decrease when the mean wind direction changes its angle to be more perpendicular to the pair.

**RC12** *line 21 ==> average size of [new]? Wind farms*

    **AC** We clarified this as follows:

> p.1, ll. 23f: Due to the increased size of the newly installed wind farms, the average size of offshore wind farms rose to 621 MW (Ramírez et al., 2020).

**RC13** *line 44: "Furthermore, the variance of the wind velocity and the mean velocity turned out to be important parameters in the modelling set up." ==> Please clarify to which modeling setup you are referring.*

    **AC** Thank you for this note. We clarified the text as follows:

> p. 2, ll. 47f: In an LES study by Lukassen et al. (2018), space-time correlations of velocity fluctuations within a wind farm with periodic boundary conditions (modelling a periodic array of wind turbines) were analysed and modelled analytically. Velocity fluctuations, which are directly related to power output fluctuations, showed pronounced space-time correlations. Furthermore, the variance of the wind velocity and the mean velocity turned out to be important parameters in the space-time correlation model.

**RC14** *line 58: "includes unstable inflow conditions," ==> please be more precise. LES and wind tunnel data also include unsteady effects. However, typically LES and wind tunnel have constant wind directions, while wind direction varies in field data, etc.*

    **AC** We agree and revised the text as follows:

> p. 3, ll. 61f: In contrast to the wind tunnel measurements by Bossuyt et al. (2017) and the LES analysis by Lukassen et al. (2018) mentioned above, the data set processed here includes unstable inflow conditions (varying wind speeds and wind directions), dynamically operating wind turbines as well as changing flow conditions within the wind farm.

**RC15** *line 70: "relevant wind turbine statistics" ==> statistics on the power production of the turbines not statistics of the turbines themselves.*

    **AC** Revised as follows:

> p.3, ll. 76f: With this and the results from the LES analysis by Lukassen et al. (2018), we identify relevant wind turbine power output statistics that influence the correlation.

**RC16** *line 95: Please be more precise on what wind direction fluctuations you refer to. Fluctuations around some mean value? The average wind direction does namely matter.*

    **AC** Yes, that is correct. Here, wind direction fluctuations around a mean wind direction are considered. We added this in the text as follows:

> p.2, ll. 54f: Dai et al. (2017) analysed 1 Hz wind farm SCADA data concerning the influence of wind speed fluctuations around a mean velocity and wind direction fluctuations around a mean wind direction on the wind turbine power output fluctuations of single wind turbines.

**RC17** *line 109-110: "All these factors multiply to an order of unpredictable variability" ==> please clarify your formulation.*

    **AC** We revised the text as follows:

> p. 5, ll. 115f: The combination of all these factors causes highly dynamic flow conditions and thus, an unpredictable variability.

**RC18** *line 135: As mentioned before, these effects have a limited ==> sentence is not complete.*

    **AC** Thank you for noting this. We completed the sentence as follows:

> p. 6, ll. 142f: As mentioned before, these effects have a limited effect on the power output fluctuations of the wind turbines.

**RC19** *line 186: why do you have the reference of $13ms^{-1}$?*

    **AC** The value was empirically chosen based on the power curve of the wind turbines considered here. Full load is reached at $12.5\,\mathrm{ms}^{-1}$; thus, $13\,\mathrm{ms}^{-1}$ (including tolerance) was chosen as we only consider partial load. We clarified this in the text as follows:

> p. 7, ll. 187f: Next, the correlation curves with the normalised lag $\tau_{norm,intv}$ are discretised using a histogram with a reference time lag of
>
> $$\tau_{norm} = \tau \cdot \frac{U_{max}}{x_{AB,mean}} \qquad (1)$$

where $\tau$ is the time lag (0 s to 300 s), $U_{max}$ is an artificially introduced velocity that has to be at least equal to the maximum possible wind speed to fit all normalised curves ($U_{max} = 13$ ms$^{-1}$ for this case). This value is based on the wind turbine power curve characteristics, including a tolerance as the wind turbines considered here reach their rated power at 12.5 ms$^{-1}$.

**RC20** *Figure 2 /3: For certain wind directions you have less measurements. The gap seems to be at slightly different wind directions in figure 2 and 3 (left / right of the 0/360 degree line). I guess it has to do with the way you defined your measurement intervals, but please make sure this is clarified to the reader.*

    **AC** Thank you for pointing this out. We revised the text as follows:

> p. 8, ll. 230f: Due to the varying data availability per wind direction and the applied data filtering (see sec. **??**), the average correlation curve per bin is based on a different number of data. It turns out that after filtering, no data is available for the bins around 330° to 10°.

**RC21** *line 224: "are ideal compared to free field measurements" ==> I agree. Can you clarify to the reader what specific aspects you believe are the main reason for the observed differences between field data and LES and wind tunnel data?*

    **AC** We believe the main reasons for this are the wind direction and wind speed changes and the individual operation of the wind turbines (yawing, limited power, shut off). To further clarify this, we added the following text:

> p. 9, ll. 237f: These dynamics are most likely caused by the wind direction and wind speed changes and the individual operation of the wind turbines (yawing, limited power, shut off). Even though the correlation curves were adapted to the average wind speed per interval, the wind speeds were just an assessment and could change during the interval. Also, the wind direction is averaged over the whole wind farm, which means certain intervals could include data from wind turbine pairs facing a slightly different direction. Further, we only consider the intervals of wind turbine pairs that fit the data filter; however, other wind turbines could be yawing at the same time or start pitching. Thus, the flow within the wind farm could still be influenced by these wind turbines.

**RC22** *line 337: "more defined averaged correlations" ==> please clarify*

    **AC** Here we would like to point out that the peaks of the average correlation curves for 270° are more narrow, thus more defined. We revised the text as follows:

> p. 16, ll. 364f: In general, we found that the averaged correlation curves of power output fluctuations for 270° with a maximum correlation coefficient of 0.21 have more defined (narrower) peaks compared to those of the averaged correlation curves for 90° with a maximum correlation coefficient of 0.16.

**RC23** *line 339: "introduced parameters" ==> please clarify what parameters you refer to.*

    **AC** We revised the text as follows:

> p. 17, ll. 366f: Further, the standard deviation of the power output fluctuations of the wind turbines in a pair was larger for 270° than for 90°.

**RC24** *line 343: "more defined" ==> please clarify what you mean*

    **AC** Similar to **RC22**, we revised the text as follows:

> p. 17, ll. 372f: We found different correlation curves for the rows of the wind farm, becoming more defined (more narrow peaks) towards the back of the wind farm.

**RC25** *line 349: "turbine pairs in the same row are affected by the same flow conditions," ==> please clarify*

   **AC** The flow conditions within the wind farm are dynamically changing due to the individual control and operation of the wind turbines. We clarified this in the text as follows:

> p. 17, ll. 379f: As mentioned before, the flow throughout the wind farm is highly variable due to the individual control and operation of the wind turbines. This means that not all wind turbine pairs in the same row are affected by the same flow conditions, as upstream wind turbines could be turned off, could be yawing or could be pitching. This further means that they show different correlation curves and should be sorted into different correlation states.

**RC26** *line 354: "clearly distinguishable parameters" ==> What parameters do you mean?*

   **AC** We revised the text as follows to clarify our focus:

> p. 18, ll. 385f: The clustering showed similar results for the wind directions 90° and 270°. The clusters had distinguishable values in the standard deviation of the power output fluctuations and in the normalised power output differences of the wind turbines, which were directly related to the average correlation curve per cluster.

**RC27** *line 361: "The remaining clusters were not as significant as the other but also showed" ==> In this sentence it is not so clear to which clusters you refer.*

   **AC** Thank you for noting this, we clarified this in the text as follows:

> p. 18, ll. 395f: Clusters 2, 3 and 4 were not as significant as Clusters 1 and 5 and showed distinguishable correlation curves with their peaks ranging from 0.14 to 0.22 for 90° and from 0.2 to 0.31 for 270°.

**Authors response to comments from referee #2 - Mark Kelly**

**RC1** *The draft has been improved with revision; however, some issues still remain. These are pointed out below, with author comments (AC) addressed first, and later point-wise comments with line numbers referring to the file that showed revision/changes ("...ATC1.pdf").*

*There are also numerous linguistic errors which need to be corrected (most of them new). As in the previous review, I again suggest asking someone with native-level English proficiency to proof-read the (updated) draft; they are too numerous to note individually here.*

*E.g., on l.6: "in free field" is not proper English; on l.8 the word 'of' is missing after "eight".*

*Before getting to specific points line-by-line, I'll respond to the author comments (ACs) that addressed my previous reviewer comments (RCs), and then I add some general comments.*

    **AC** Thank you again for your feedback. We hope the professional proofreading improved our manuscript. Please see the track of changes for the overall revision. In the following, we will address your comments specifically.

Comments on previous review

**RC3** *AC11: It is an improvement to mention the "reconstructed" aspect of wind speed, but to be open/clear, why not include your AC statement "details on the reconstruction are not available" in the text? This and/or the unknown transfer function should be mentioned; e.g. the latter can affect the direction as well as the speed.*

    **AC** Thank your for noting this, we added this information to the text as follows:

> p. 3, ll. 90f: The reconstructed wind speed $U$ is not directly measured but provided as a variable that results from the measured power and control variables of the wind turbine (details on the reconstruction of $U$ are not available).

**RC4** *AC12: Yaw misalignment isn't only due to "false...measurements, calibrations, or sensor installation"–especially if there is a transfer function used for nacelle-mounted 2d-anemometers. Your addition on l.106-8 helps to allay this issue.*

    **AC** Yes, we agree and revised the text as follows:

> p. 3, ll. 91f: Due to this, $U$ is considered as an approximated and idealised value that does not include the wind speed independent power reduction, e.g. by a yaw misalignment of the wind turbine.

**RC5** *AC15: Your response about yaw misalignment threshold and rate per 10-minutes is reasonable, but you have not included this in the revised text.*

    **AC** We clarified our description of the selection of the wind direction as follows:

p. 6, ll. 138f: Since this data filter only applies to the average wind direction $\Phi_{av}$, individual wind turbines might have a slightly deviating relative wind direction for this specific time interval. This deviation could be caused by a false wind direction measurement, a yawing process that has taken place asynchronously to the majority of other wind turbines or a wind direction deviation due to local changes over the area of the wind farm. This means there is no threshold for yaw misalignment within the 600 s intervals.

**RC6** *AC17: How is your approach "similar to" Taylor's, but not actually simply assuming it? Again, it appears you've assumed it over the entire range of $\tau$ (and $\tau$norm); this can become problematic for small enough U (large lags).*

    **AC** We agree that Taylor's frozen eddy hypothesis could be problematic. However, we do not assume frozen eddies here. Turbulence structures travel downstream, but they are subject to change when doing so. We just use the mean velocity as an advection velocity for turbulent structures, which is similar to Taylor's hypothesis. In our case, small velocities (causing large lags) are not a problem as the power output fluctuations would not be correlated anymore in this case. To clarify this, we revised the text as follows:

p. 7, ll. 175f: Similar to Taylor's hypothesis (Taylor, 1938), we assume that the wind structures responsible for the power output fluctuations measured at an upstream wind turbine A that travel a certain distance to the downstream wind turbine B with an advection speed that is similar to the average wind speed over that distance. But in contrast to Taylor's hypothesis, we do not assume frozen eddies but expect wind structures to change and thus decorrelate while travelling downstream. Further, as we have no access to the average wind speed over the distance between wind turbines A and B, we use the average wind speed measured at wind turbine B as a reference.

**RC7** *AC18 and l.189-205: you have used $\tau$norm before it is defined in (4); this can be quite confusing for the reader, particularly if they have not read this before. Also, why is the second 'normalization' (4) done? If Umax = 13m/s always, and xAB; mean is also a simple constant for all cases, then why normalize again? If you have done this to force the peaks closer to 1, then this should be stated. Also, how was Umax chosen–doesn't this just arbitrarily squeeze/stretch the correlation curves (as you wrote for $\tau$norm;intv)? There appears to be no physical justification for (3) and (4) together, unless perhaps you could explain what is meant by "at least equal to the maximum possible wind speed to fit all normalised curves".*

    **AC** Thank you for pointing this out. We adjusted the order of the text and further clarified the usage of $\tau_{norm}$ and $\tau_{norm,intv}$. While $\tau_{norm,intv}$ is used to shrink and stretch the correlation curves based on the average wind speed within a time interval, $\tau_{norm}$ is a reference time lag which is only created to bin $\tau_{norm,intv}$ in a histogram. $\tau_{norm}$ is thus the reference time lag used for representing all processed results. $U_{max}$ was empirically chosen based on the power curve of the here considered wind turbines. Full load is reached at $12.5 \text{ ms}^{-1}$, thus $13 \text{ ms}^{-1}$ (including tolerance) was chosen as we only consider partial load. The text in the manuscript was revised as follows:

p. 7, ll. 180f: Hence, to compare the correlations calculated for intervals with different average wind speeds and different wind turbine distances, the time lag $\tau$ is normalised for each time interval starting at $t_j$ individually

$$\tau_{norm,intv} = \tau \cdot \frac{\langle U_B(t+\tau)\rangle_{\Delta t_{300}}}{x_{AB}} \tag{2}$$

where $\tau_{norm,intv}$ is the normalised time lag, $\langle U_B(t+\tau)\rangle_{\Delta t_{300}}$ is the average reconstructed wind speed from a certain (downstream) wind turbine B for a time interval $\Delta t_{300} = 300$ s for $t$ in the discretised interval $[t_j, t_j + 299 \text{ s}]$ and a certain lag $\tau$. This means for a certain $\tau$, the averaging interval of $\langle U_B(t+\tau)\rangle_{\Delta t_{300}}$ is $[t_j + \tau, t_j + \tau + 299 \text{ s}]$. $x_{AB}$ is the distance between wind turbine A and wind turbine B.

Next, the correlation curves with the normalised lag $\tau_{norm,intv}$ are discretised using a histogram with a reference time lag of

$$\tau_{norm} = \tau \cdot \frac{U_{max}}{x_{AB,mean}} \tag{3}$$

where $\tau$ is the time lag (0 s to 300 s), $U_{max}$ is an artificially introduced velocity that has to be at least equal to the maximum possible wind speed to fit all normalised curves ($U_{max} = 13$ ms$^{-1}$ for this case). This value is based on the wind turbine power curve characteristics, including a tolerance as the wind turbines considered here reach their rated power at 12.5 ms$^{-1}$. $x_{AB,mean}$ is the average distance between wind turbine A and wind turbine B of the considered wind turbine pairs. Note that $\tau_{norm,intv}$ is used for stretching and shrinking of the correlation curves. $\tau_{norm}$ is only a reference time lag that is only created for binning of the stretched or shrunk correlations and does not change the correlation curves.

**RC8** *AC20: The response statement "the temporal autocorrelation of a wind turbine decorrelates in the considered time intervals of 300 s" does not make sense. Do you perhaps mean that the the correlation decreases to effectively 0 as lag ($\tau$) approaches 300s?*

    **AC** Yes, exactly. We meant that the correlation decreases to 0 within the considered time lag of 300 s.

**RC9** *AC23: If you are to insist on using the term "filtering" in place of 'data selection' or similar—knowing that WES is not a data science journal, but a wind energy journal where you are also mentioning turbulence—then you should at least include the word 'data' before it. Further, I strongly recommend section 2.1 to be renamed "Data selection and filtering" or similar—again, spectral filtering is commonly used when dealing with this kind of data in wind energy (particularly turbulence), especially when mentioning different intervals (e.g. 600s).*

    **AC** Thank you very much, for this valuable suggestion. We renamed the section 2.1 to "Data selection and filtering" and changed the term 'filtering' to 'data filtering' in the text. For the latte, please see the track of changes.

Underline{General comment}

**RC10** *The labelling of peak correlations (between power fluctuations for turbine pairs) as "correlation states" is contentious, since 'states' implies different physical scenarios or flow/operational-regimes—particularly if you have not described anything like the latter. If the 'states' are basically different groups of peak correlations (magnitudes), then why not call them that? This is safer, because for different flow regimes having larger/smaller turbulence length scales (and/or other farms having different spacing, surfaces, or even hub height), then the magnitudes or groups could be quite different. Also, in the conclusion it should be mentioned how/why seemingly insignificant peak correlations (e.g. 0.2 or less) are meaningful (compared to commonly understood statistical significance being $\rho_{AB} > 0.5$); i.e. the relative values are significant given the 'noisy' turbulent flow, in addition to values consistent with others' LES results.*

    **AC** We use the term states indeed to describe a group of certain correlations or correlation curves. However, these states are identified especially for this wind farm and the here considered time period. To emphasize this, we adjusted the definition of correlation states in the text as follows:

p. 3, ll. 64f: In this paper, we investigate the influencing factors on the correlation of power output fluctuations of wind turbine pairs and introduce parameters to distinguish different correlation curves, herein called correlation states. A state defines a group of similar correlation curves. Note that the states found here refer to this specific wind farm and the considered time period.

Hence, our understanding of the term correlation states is not a 'global' but a 'local' one. We do not expect to find exactly the same correlation states for other time periods or other wind farms. However, the process of the identification of correlation states stays the same and applies to any other condition.

The evaluation of the peak correlations in the conclusion was revised as follows:

> p. 17, ll. 368f: In the context of the considered highly varying flow conditions, peak correlations around 0.21 or 0.16 are still considered significant. The cause for these relatively low peak correlations lies in the varying flow conditions or noisiness of the flow within the wind farm.

Specific comments

**RC30** *l.5, 7: the challenge is spatially variable flow, not just "highly" variable flow.*

    **AC** You are right, it is spatial variable but also temporally variable. To shorten the expression we stick to highly variable. We revised the text as follows:

> p.1, ll. 4f: Here, we present an approach to investigate space-time correlations of power output fluctuations of streamwise-aligned wind turbine pairs based on high-resolution SCADA data. The proposed approach overcomes the challenge of spatially variable and temporally variable flow conditions within the wind farm.

**RC31** *l.10-11: "decrease towards spanwise pairs" doesn't quite make sense. If the correlations decrease with angle between mean wind direction and pair separation vector, why not write that?*

    **AC** Thank you for noting this. We revised the text as follows:

> p. 1, ll. 8f: First, we asses the effect of the wind direction on the correlations of power output fluctuations of wind turbine pairs. We show that the correlations are highest for the streamwise-aligned wind turbine pairs and decrease when the mean wind direction changes its angle to be more perpendicular to the pair.

**RC32** *l.13-17: "the correlation of streamwise aligned wind turbine pairs" should be 'power correlations between streamwise-aligned wind turbine pairs'.*

    **AC** We agree and revised the text as follows:

> p. 1, ll. 11f: Further, we show that the correlations for streamwise-aligned wind turbine pairs depend on the location of the wind turbines within the wind farm and on their inflow conditions (free stream or wake).

**RC33** *l.18: sorting is accomplished by a "k-means clustering algorithm", not "clustering algorithm k-means" (e.g. Likas, Vlassis, & Verbeek 2003).*

    **AC** Thank you for pointing this out. We revised the text accordingly:

> p. 1, ll. 15f: For this, we employ the data-driven $k$-means clustering algorithm to cluster the standard deviations of the power output fluctuations of the wind turbines and the normalised power difference of the wind turbines in a pair.

**RC34** *.19-20: the sentence "These groups are here referred to as correlation states." is not needed in an abstract.*

    **AC** We agree and erased this sentence in the abstract:

> p. 1, ll. 12f: Our primary result is that the standard deviations of the power output fluctuations and the normalised power difference of the wind turbines in a pair can characterise the correlations of power output fluctuations of streamwise-aligned wind turbine pairs.

**RC35** *l.20 repeats l.7-8.*

    **AC** The last sentences of the abstract were adjusted as follows:

> p. 1, ll. 15f: For this, we employ the data-driven *k*-means clustering algorithm to cluster the standard deviations of the power output fluctuations of the wind turbines and the normalised power difference of the wind turbines in a pair. Thereby, wind turbine pairs with similar power output fluctuation correlations are clustered independently from their location. With this, we account for the highly variable flow conditions inside a wind farm, which unpredictably influence the correlations.

**RC36** *l.18-22: "these parameters" is repeated three times; the final point is also somewhat of a repetition of l.16-17...The abstract can be cleaned up (there are more English errors in it as well).*

    **AC** Thank you for pointing this out. We totally agree and revised the whole abstract. Please see the track of changes for details.

**RC37** *l.71: Use of "correlation curves" here to mean "states" is ambiguous and confusing; however, in l.194 and after it is used reasonably, to refer to the actual $R(\tau)$ curves. Here in this context of "states" you are really referring to the peak correlation, as seen later in e.g. Figs.7-8.*

    **AC** Thank you for noting this. We agree and revised the text as follows:

> p. 3, ll. 67f: The parameters introduced to characterise correlation curves are then evaluated with a data-driven clustering algorithm to group the data according to the underlying correlation curves.

**RC38** *l.151: sentence is not finished.*

    **AC** Thank you for noting this. The sentence was finished as follows:

> p. 6, ll. 142f: As mentioned before, these effects have a limited effect on the power output fluctuations of the wind turbines.

**RC39** *l.196: "noted as" –> 'denoted by'*

    **AC** Revised.

**RC40** *l.212 and elsewhere: "dependency" should be 'dependence'*

    **AC** Revised. Please find all adjustments in the track of changes.

**RC41** *l.242: how does $\tau_{norm} > 1$ have any meaning, if $\tau_{norm}$ is arbitrary due to its definition via the artificial Umax?*

    **AC** Our answer to **RC7** most likely already helps to clarify this. As $\tau_{norm}$ is only used to bin the shrunk or stretched correlation curves, $\tau_{norm} = 1$ still represents the case when the advection speed of the fluctuations equals the wind speed measured at wind turbine B. The binning only slightly affects the peaks of the correlations due to the averaging of the values inside the respective bins, which also applies to the bin around $\tau_{norm} = 1$.

**RC42** *l.381: exactly which standard deviation?*

    **AC** We clarified this sentences as follows:
* * *
p. 17, ll. 366f: Further, the standard deviation of the power output fluctuations of the wind turbines in a pair was larger for 270° than for 90°.
* * *
**References**

Bossuyt, J., Howland, M. F., Meneveau, C., and Meyers, J.: Measurement of unsteady loading and power output variability in a micro wind farm model in a wind tunnel., Experiments in Fluids, 58, 1–17, https://doi.org/10.1007/s00348-016-2278-6, 2017.

Dai, J., Cao, J., Liu, D., Wen, L., and Long, X.: Power fluctuation evaluation of large-scale wind turbines based on SCADA data, IET Renewable Power Generation, 11, 395–402, https://doi.org/10.1049/iet-rpg.2016.0124, 2017.

Ester, M., Kriegel, H.-P., Sander, J., and Xu, X.: A Density-Based Algorithm for Discovering Clusters in Large Spatial Databases with Noise, in: Proceedings of the Second International Conference on Knowledge Discovery and Data Mining, KDD'96, p. 226–231, AAAI Press, 1996.

Kaufman, L. and Rousseeuw, P.: Partitioning Around Medoids (Program PAM), chap. 2, pp. 68–125, John Wiley & Sons, Ltd, https://doi.org/10.1002/9780470316801.ch2, 2008.

Lukassen, L. J., Stevens, R. J. A. M., Meneveau, C., and Wilczek, M.: Modeling space-time correlations of velocity fluctuations in wind farms, Wind Energy, 21, 474–487, https://doi.org/10.1002/we.2172, 2018.

Ramírez, L., Fraile, D., and Brindley, G.: Offshore Wind in Europe - Key trends and statistics 2019, https://windeurope.org/wp-content/uploads/files/about-wind/statistics/WindEurope-Annual-Offshore-Statistics-2019.pdf, last access 21.01.2021, 2020.

Taylor, G. I.: The Spectrum of Turbulence, Proceedings of the Royal Society A: Mathematical, Physical and Engineering Sciences, 164, 476–490, https://doi.org/10.1098/rspa.1938.0032, https://royalsocietypublishing.org/doi/pdf/10.1098/rspa.1938.0032https://royalsocietypublishing.org/doi/10.1098/rspa.1938.0032, 1938.